# MC-HNN: Learning Latent Structural Semantics and High-Rank Representations for Hypergraph Neural Networks

Shuyang Fang [1]  Yuqin Huang [1]  Zelong Yang [2]  Yintao Cai [1]  Xiaoping Min [1]

## Abstract

Hypergraph Neural Networks (HNNs) have emerged as powerful tools for modeling complex high-order correlations. Most existing HNNs adhere to a two-stage message passing paradigm, where node feature propagation is mediated by hyperedges. In this paper, we analyze two structural limitations of this paradigm, which we term rank collapse and hyperedge semantic dependency. To address these challenges, we propose the Multi-Channel Hypergraph Neural Network (MC-HNN). We design a multi-channel message passing mechanism to maintain high-rank representations, while simultaneously introducing a latent hyperedge type encoding mechanism to inject an independent degree of freedom into hyperedge representations. Our analysis and experiments suggest that MC-HNN alleviates these bottlenecks and achieves strong empirical performance. Our code is available at https://github.com/Kssits/MC-HNN.

## 1. Introduction

Graph Neural Networks (GNNs) (Scarselli et al., 2008; Bruna et al., 2014; Defferrard et al., 2016; Kipf & Welling, 2017; Veličković et al., 2018; Hamilton et al., 2017) have emerged as the dominant paradigm for processing graph-structured data, achieving remarkable success in domains such as social network analysis (Li & Goldwasser, 2019), bioinformatics (Yan et al., 2019), and citation networks (Zhao et al., 2021). However, traditional graph models are fundamentally limited to modeling pairwise relationships

[1] School of Informatics, State Key Laboratory of Molecular Vaccinology and Molecular Diagnostics, National Institute of Diagnostics and Vaccine Development in Infectious Diseases, Xiamen University, Xiamen, China [2] Institute of Artificial Intelligence, Xiamen University, Xiamen, China . Correspondence to: Xiaoping Min <mxp@xmu.edu.cn>.

*Proceedings of the 43rd International Conference on Machine Learning*, Seoul, South Korea. PMLR 306, 2026. Copyright 2026 by the author(s).

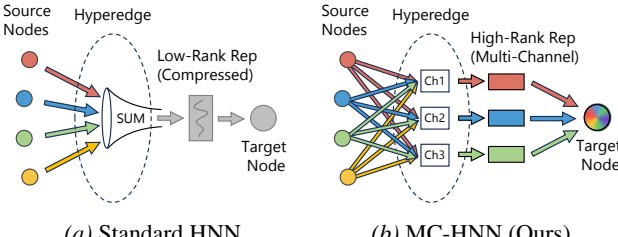

*(a)* Standard HNN          *(b)* MC-HNN (Ours)

*Figure 1.* Conceptual comparison of message passing paradigms. (a) In **Standard HNNs**, dense connectivity forces diverse node features through a single aggregation bottleneck (SUM), resulting in a compressed, low-rank hyperedge representation that causes information loss. (b) Our proposed **MC-HNN** employs a multi-channel aggregation mechanism. This preserves a high-rank hyperedge representation, allowing the target node to adaptively integrate rich, distinct semantic features from different channels.

between entities, often failing to capture the high-order interactions prevalent in real-world scenarios. For instance, a research paper is typically co-authored by multiple researchers, and a biochemical reaction involves the synergistic action of various proteins. Hypergraphs, by introducing "hyperedges" that connect an arbitrary number of nodes, naturally align with the imperative to model such complex correlations. Empirical evidence (Veldt et al., 2022; Schaub et al., 2021; Battiston et al., 2020; Lambiotte et al., 2019; Lee et al., 2021) suggests that Hypergraph Neural Networks (HNNs) (Chien et al., 2022; Huang & Yang, 2021; Wang et al., 2023; Li et al., 2025) significantly outperform traditional graph baselines in extracting high-order semantics and representing complex topological structures.

Currently, HNNs predominantly adopt a two-stage message passing paradigm: messages are first aggregated from nodes to hyperedges, and subsequently distributed back to nodes, indirectly enabling node-to-node interaction. However, a single hyperedge often encapsulates a large number of nodes, while each node simultaneously participates in multiple hyperedges. Such dense connectivity causes a node to receive an overwhelming amount of neighborhood information from omnipresent sources within a single message-passing iteration. Such a mechanism renders hypergraph learning particularly susceptible to over-squashing and over-smoothing (Xu et al., 2019; Topping et al., 2022;

Alon & Yahav, 2021). As the number of propagation layers increases, node representations inevitably become homogenized and indistinguishable.

Existing approaches (Ding et al., 2020; Huang & Yang, 2021; Chien et al., 2022; Wang et al., 2023) primarily focus on refining message passing by optimizing aggregation functions or propagation mechanisms. More recently, other works have sought to enhance model expressivity by expanding receptive fields (Xie et al., 2025) or harmonizing high- and low-frequency signals (Li et al., 2025).

We revisit the standard HNN propagation paradigm from the perspective of representation capacity and hyperedge dependency. In this work, we study two structural bottlenecks in the standard HNN paradigm. First, as shown in Figure 1(a), over-squashing in hypergraphs originates locally during the node-to-hyperedge aggregation phase, rather than solely arising from long-range propagation as observed in standard graphs; we formally define this phenomenon as rank collapse. Second, hyperedge representations are derived exclusively from node features, lacking the intrinsic degrees of freedom to capture latent semantics independent of their constituents. This limitation undermines the structural significance of hyperedges, a strictly coupled state we term hypergraph semantic dependency.

Based on these insights, we propose the **Multi-Channel Hypergraph Neural Network (MC-HNN)**. Motivated by the imperative to preserve information capacity, we design a multi-channel message passing framework capable of sustaining high-rank representations, as shown in Figure 1(b). Simultaneously, we incorporate a latent hyperedge type encoding, granting hyperedges additional degrees of freedom to capture emergent semantics. Furthermore, a dynamic gating mechanism is employed, enabling nodes to adaptively select and integrate information from relevant channels for effective updates.

Our contributions are summarized as follows:

- We analyze two structural bottlenecks in standard HNN propagation: rank collapse during hyperedge aggregation and the dependency of hyperedge representations on aggregated node features.

- We propose MC-HNN, a framework with multi-channel message passing and latent hyperedge type encoding, and discuss how these components increase rank capacity and introduce node-independent degrees of freedom.

- We conduct experiments on diverse hypergraph benchmarks, showing that MC-HNN achieves strong performance.

## 2. Related Work

### 2.1. Hypergraph Neural Networks

Hypergraph Neural Networks (HNNs) generalize Graph Neural Networks (GNNs) to model high-order relationships among entities. Early approaches primarily adapted spectral theory to hypergraphs. HGNN (Feng et al., 2019) pioneered this direction by employing clique expansion to define a hypergraph Laplacian for spectral convolution. Similarly, HyperGCN (Yadati et al., 2019) approximates hyperedges as simple edges by selecting representative node pairs based on total variation, though this reduction inevitably entails structural information loss.

Moving beyond spectral approximations, spatial message passing has become the dominant paradigm. UniGNN (Huang & Yang, 2021) formalized a unified framework based on the incidence graph, establishing the standard two-stage propagation: node-to-hyperedge aggregation followed by hyperedge-to-node updates. Subsequent works have focused on refining these aggregation functions. HNHN (Dong et al., 2020) introduces flexible normalization and distinct weight matrices for both stages. AllSet (Chien et al., 2022) generalizes the aggregation process into two learnable multiset functions, instantiated via permutation-invariant structures like Deep Sets or Set Transformers. More recently, ED-HNN (Wang et al., 2023) leverages hypergraph diffusion theory to generate discriminatory messages, effectively addressing heterophily. Alternatively, Hyper-MLP (Tang et al., 2024) eschews explicit message passing in favor of a Laplacian smoothing regularizer to encode structural priors efficiently. KHGNN (Xie et al., 2025) extends the receptive field by designing a bipartite nested convolution for $K$-hop communication.

Despite these methodological innovations, most existing HNNs adhere to a monolithic message passing pipeline. As discussed in the Introduction, irreversible information loss is inevitably incurred when node information is aggregated to hyperedges.

### 2.2. Factorized and Multi-Branch Graph Learning

Factorization and multi-branch architectures have been explored in standard GNNs to disentangle complex underlying factors. DisenGCN (Ma et al., 2019) proposes a neighborhood routing mechanism, assigning neighbors to distinct latent channels to learn disentangled node representations. M2M-GNN (Liang et al., 2024) adopts a multiset-to-multiset paradigm, partitioning neighbors into different subsets to prevent information mixing in heterophilic graphs.

This concept has recently emerged in hypergraph learning, predominantly in application-specific domains. In spatio-temporal analysis, DisenHCN (Li et al., 2022b) explicitly decomposes representations into predefined aspects—such

as location, time, and activity—propagating information independently for each aspect. In session-based recommendation, HIDE (Li et al., 2022a) splits item embeddings into multiple intent partitions, processing them via parallel encoders to capture diverse user interests.

Recent studies have further explored disentangled representations for high-order structures and hypergraphs. HSDN (Hu et al., 2022) constructs hyperedges from graph structure units and learns disentangled high-order structural semantics. Natural-HNN (Lee et al., 2026) further studies hyperedge-level disentanglement via a category-theoretic naturality criterion, demonstrating its effectiveness on cancer subtype classification over genetic pathway hypergraphs.

These studies further demonstrate the relevance of factorized and disentangled representations for high-order relational learning, while primarily emphasizing semantic factorization and interpretability.

# 3. Preliminaries and Problem Formulation

## 3.1. Preliminaries

**Notation.** Let $\mathcal{G} = (\mathcal{V}, \mathcal{E}, \mathbf{X})$ denote a hypergraph, where $\mathcal{V} = \{v_1, \ldots, v_N\}$ is the set of $N$ nodes, and $\mathcal{E} = \{e_1, \ldots, e_M\}$ is the set of $M$ hyperedges. $\mathbf{X} \in \mathbb{R}^{N \times F}$ represents the initial node feature matrix, with $\mathbf{x}_i \in \mathbb{R}^F$ denoting the feature vector of node $v_i$. Each hyperedge $e_j \in \mathcal{E}$ represents a subset of $\mathcal{V}$. The topological structure is encoded by an incidence matrix $\mathbf{H} \in \{0, 1\}^{N \times M}$, where $\mathbf{H}_{ij} = 1$ if node $v_i \in e_j$, and 0 otherwise. Based on $\mathbf{H}$, for any node $v_i$ and hyperedge $e_j$, we define the incident hyperedge set $\mathcal{E}_i = \{e_j \in \mathcal{E} \mid \mathbf{H}_{ij} = 1\}$ and the constituent node set $\mathcal{V}_j = \{v_i \in \mathcal{V} \mid \mathbf{H}_{ij} = 1\}$, respectively.

**The Unified Paradigm of Hypergraph Neural Networks.** While numerous HNN variants exist, most state-of-the-art methods adhere to a unified aggregation-and-transformation paradigm. The node representation update at the $l$-th layer can be formalized as a two-stage process:

$$\mathbf{h}_e^{(l)} = \mathcal{A}_{\mathcal{V} \to \mathcal{E}} \left( \left\{ \psi \left( \mathbf{x}_u^{(l)} \right) : u \in \mathcal{V}_e \right\}; \theta_1 \right), \quad (1)$$

$$\mathbf{x}_v^{(l+1)} = \mathcal{U} \left( \mathbf{x}_v^{(l)}, \mathcal{A}_{\mathcal{E} \to \mathcal{V}} \left( \left\{ \varphi \left( \mathbf{h}_e^{(l)}, \mathbf{x}_v^{(l)} \right) : e \in \mathcal{E}_v \right\}; \theta_2 \right) \right). \quad (2)$$

In the first stage (Eq. 1), node information is aggregated into hyperedge representations. Here, $\mathbf{x}_u^{(l)}$ denotes the features of node $u$ at layer $l$, and $\mathcal{A}_{\mathcal{V} \to \mathcal{E}}$ serves as the node-to-hyperedge aggregator (e.g., sum, mean, or attention). The function $\psi(\cdot)$ applies a feature transformation, typically parameterized by a linear projection or MLP, prior to aggregation.

In the second stage (Eq. 2), hyperedge messages are distributed back to nodes. The function $\varphi(\cdot, \cdot)$ constructs the

message flowing from hyperedge $e$ to node $v$. In models like UniGNN (Huang & Yang, 2021) and AllSet (Chien et al., 2022), $\varphi$ typically depends solely on $\mathbf{h}_e^{(l)}$, treating all nodes within a hyperedge equally. Conversely, approaches like ED-HNN (Wang et al., 2023) incorporate both $\mathbf{h}_e^{(l)}$ and $\mathbf{x}_v^{(l)}$ into $\varphi$, allowing for anisotropic message passing that adapts to heterophilic connections. Finally, $\mathcal{A}_{\mathcal{E} \to \mathcal{V}}$ aggregates these messages, and $\mathcal{U}$ updates the node embedding for the next layer.

## 3.2. Theoretical Limitations of the Unified Paradigm

**Rank Collapse in Hyperedge Aggregation.** In the first stage ($\mathcal{V} \to \mathcal{E}$), the set of nodes connected by a hyperedge $e$ collectively holds rich semantic information. We formalize their joint representation as follows:

**Definition 3.1** (Local Incidence Feature Matrix). For a hyperedge $e$ incident to $k$ nodes $\mathcal{V}_e = \{v_1, \ldots, v_k\}$, the local feature matrix $\mathbf{X}_{\mathcal{V}_e} \in \mathbb{R}^{F \times k}$ is constructed by stacking the feature vectors of all incident nodes: $\mathbf{X}_{\mathcal{V}_e} = [\mathbf{x}_1^\top, \ldots, \mathbf{x}_k^\top]$.

In high-dimensional feature spaces (typically $F \gg k$), node features are typically in general position, meaning they are linearly independent almost surely despite potential semantic similarities. Consequently, the local feature matrix $\mathbf{X}_{\mathcal{V}_e}$ has a generic rank of $\text{rank}(\mathbf{X}_{\mathcal{V}_e}) = k$. This rank reflects the theoretical capacity required to fully preserve individual node information. However, standard aggregation functions (e.g., mean, sum, attention) compress this matrix into a single vector. We characterize this structural limitation as follows:

**Proposition 3.2** (Rank Collapse). *Given a hyperedge $e$ with a local feature matrix $\mathbf{X}_{\mathcal{V}_e}$ having rank $r$ (where typically $r = k$, and strictly $r > 1$), aggregating this matrix into a single vector representation $\mathbf{h}_e \in \mathbb{R}^F$ via strictly contractive operations enforces a hard constraint $\text{rank}(\mathbf{h}_e) \leq 1$. This induces an irreversible information bottleneck with a dimension loss of $\Delta = r - 1$. Consequently, the resulting $\mathbf{h}_e$ resides in a rank-1 subspace, limiting the capacity of this intermediate representation to preserve multiple independent directions of variation.*

See Appendix A for the detailed proof.

This rank collapse implies that $\mathbf{h}_e$ acts as an "over-squashed" summary. When this low-fidelity signal is broadcast back to nodes, it lacks the structural capacity to support complex, node-specific updates, leading to the homogenization of representations.

**Hyperedge Semantic Dependency.** Beyond capacity loss, the unified paradigm suffers from a fundamental structural limitation regarding the information source. We formally define this as hyperedge semantic dependency:

**Definition 3.3** (Hyperedge Semantic Dependency). A hyperedge representation $\mathbf{h}_e$ exhibits semantic dependency if it is strictly functionally determined by its constituent nodes, i.e., $\mathbf{h}_e = \phi(\mathbf{X}_{\mathcal{V}_e})$ for some deterministic function $\phi$. This implies that $\mathbf{h}_e$ is conditionally independent of any latent external semantics $\mathbf{z}_e$ given the nodes:

$$P(\mathbf{h}_e|\mathbf{X}_{\mathcal{V}_e}, \mathbf{z}_e) = P(\mathbf{h}_e|\mathbf{X}_{\mathcal{V}_e}). \qquad (3)$$

Standard HNNs strictly adhere to this definition, as their hyperedge representations are derived solely from node aggregations. This rigid dependency imposes two critical limitations. Primarily, the model becomes susceptible to noise amplification. Without independent correction mechanisms, $\mathbf{h}_e$ inevitably aggregates and broadcasts interfering signals from heterophilic neighbors (or noisy features), triggering error cascading during the subsequent $\mathcal{E} \rightarrow \mathcal{V}$ distribution. Furthermore, the model suffers from a loss of emergent semantics. Real-world hyperedges often embody intrinsic properties $\mathbf{z}_e$ (e.g., a specific research topic) that are not fully explicable by the summation of parts. Under the semantic dependency constraint, the model mathematically lacks the degrees of freedom to approximate any latent factors $\mathbf{z}_e$ that lie orthogonal to the node feature span.

## 4. Method

In this section, we propose the **Multi-Channel Hypergraph Neural Network (MC-HNN)** to resolve the theoretical limitations identified above. As illustrated in Figure 2, our framework features a hierarchical message passing structure that disentangles feature propagation into distinct subspaces.

### 4.1. Latent Hyperedge Type Encoding

To address the hyperedge semantic dependency discussed in Section 3.2, we introduce a latent hyperedge type encoding mechanism. By assigning a latent type to each hyperedge, we provide an external reference frame independent of the aggregated node information. This allows the model to capture meta-information that is intrinsic to the hyperedge and statistically independent of its incident nodes.

In hypergraphs characterized by a vast number of hyperedges, directly maintaining a unique high-dimensional embedding for each hyperedge entails a parameter complexity of $\mathcal{O}(M \times d)$. This is not only parameter-inefficient but also prone to overfitting, as it treats hyperedges as isolated entities without exploiting shared semantic structures. Instead, we employ a soft coding strategy using a global codebook. Let $\mathbf{T} \in \mathbb{R}^{K \times d}$ be a learnable codebook containing $K$ latent prototypes, akin to vector quantization or prototypical learning approaches (Van Den Oord et al., 2017; Snell et al., 2017). The specific latent type encoding for a hyperedge $e$ is

generated via a weighted combination of these prototypes:

$$\tilde{\mathbf{t}}_e = \text{Softmax}(\mathbf{l}_e/\tau_s)\mathbf{T}, \qquad (4)$$

where $\tilde{\mathbf{t}}_e \in \mathbb{R}^d$ is the latent type embedding for hyperedge $e$. The vector $\mathbf{l}_e \in \mathbb{R}^K$ is a learnable logit vector specifically associated with hyperedge $e$ (retrieved from a parameter matrix $\mathbf{L} \in \mathbb{R}^{M \times K}$), and $\tau_s$ is a temperature hyperparameter controlling the sharpness of the distribution. Crucially, while $\tilde{\mathbf{t}}_e$ is updated during backpropagation, it remains functionally independent of the input node features $\mathbf{X}$, thereby injecting the necessary intrinsic degrees of freedom into the hyperedge representation.

### 4.2. Dynamic Multi-Channel Message Passing

**Multi-Channel Message Passing.** To mitigate the rank collapse, we propose a multi-channel message passing mechanism. This approach disentangles the mixed signals typically compressed in a single channel, allowing the model to preserve high-rank information by distributing it across multiple independent subspaces.

Let $C$ denote the number of channels and $d$ denote the dimension per channel. We initialize the multi-channel node representations $\mathbf{X}^{(0)} \in \mathbb{R}^{N \times (d \cdot C)}$ by projecting raw features via $\mathbf{X}^{(0)} = \text{ReLU}(\mathbf{X}\mathbf{W}_{\text{in}})$, where $\mathbf{W}_{\text{in}} \in \mathbb{R}^{F \times (d \cdot C)}$. In each layer $l$, message passing proceeds in two stages:

**Stage 1: Node-to-Hyperedge Aggregation.**

We define the hyperedge representation as a composite of aggregated neighborhood information and an intrinsic latent type. This explicitly breaks the rigid functional dependency on node features:

$$\mathbf{h}_e^{(l)} = \underbrace{\frac{1}{|\mathcal{V}_e|} \sum_{v \in \mathcal{V}_e} \mathbf{x}_v^{(l-1)}}_{\text{Neighborhood Info}} + \underbrace{\tilde{\mathbf{t}}_e'}_{\text{Latent Semantics}}, \qquad (5)$$

where $\mathbf{x}_v^{(l-1)} \in \mathbb{R}^{d \cdot C}$ is the node representation, and $\tilde{\mathbf{t}}_e' \in \mathbb{R}^{d \cdot C}$ is the latent hyperedge type encoding broadcasted across all channels, i.e., $\tilde{\mathbf{t}}_e' = [\tilde{\mathbf{t}}_e \| \ldots \| \tilde{\mathbf{t}}_e]$.

**Stage 2: Hyperedge-to-Node Broadcasting.** In this phase, we compute channel-specific importance weights to selectively retrieve information inspired by Brody et al. (2022). To reduce computational complexity, we design a dual-pathway architecture: a main pathway for high-dimensional feature transmission ($d \cdot C$) and a lightweight branch pathway for importance calculation.

In the lightweight branch, we first compress the high-dimensional multi-channel features into a lower-dimensional subspace $\mathbb{R}^d$. The channel-specific raw attention score vector $\mathbf{s}_{v,e}^{(l)} \in \mathbb{R}^C$ is computed by fusing the compressed node features and hyperedge features:

$$\mathbf{s}_{v,e}^{(l)} = \mathbf{W}_{\text{att}} \left( \left[ \mathbf{W}_V \mathbf{x}_v^{(l-1)} \| \mathbf{W}_E \mathbf{h}_e^{(l)} \right] \right), \qquad (6)$$

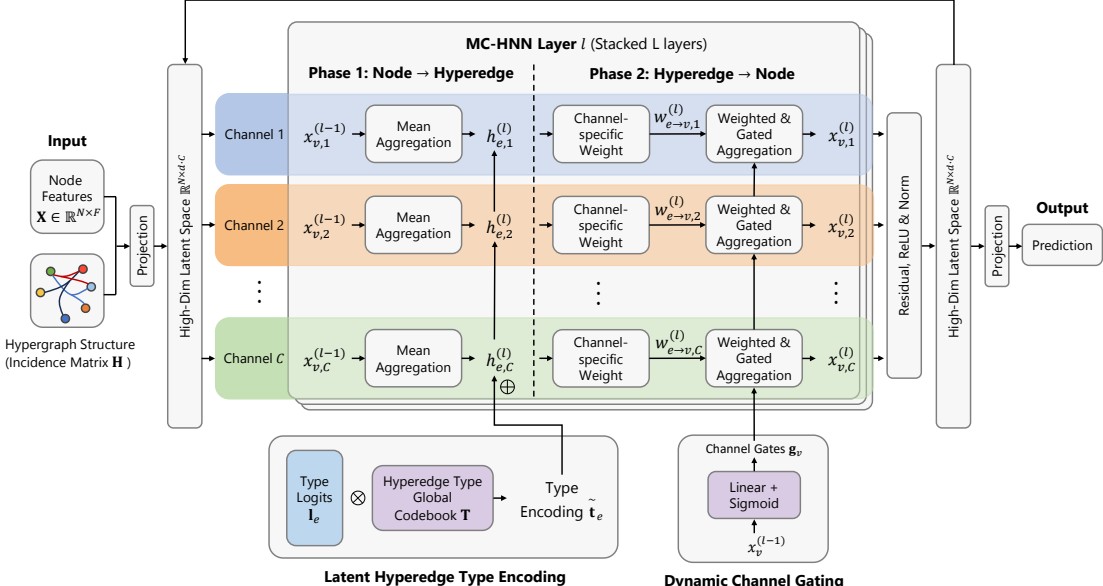

Figure 2. **The overall architecture of MC-HNN.** The model stacks $L$ layers of dynamic multi-channel message passing. In each layer, the process is divided into two phases: (1) Node-to-hyperedge Aggregation, where node features are uniformly aggregated into hyperedge representations $\mathbf{h}_e^{(l)}$; and (2) Hyperedge-to-node Broadcasting, where a channel-specific importance weight $w_{e\to v}^{(l)}$ is computed by fusing the hyperedge feature with the latent hyperedge type encoding $\tilde{\mathbf{t}}_e$. Finally, a dynamic gating mechanism $\mathbf{g}_v$ adaptively modulates the information flow for each channel. The layer update utilizes an initial residual connection to preserve raw feature information.

where $\|$ denotes concatenation. The projection matrices $\mathbf{W}_V, \mathbf{W}_E \in \mathbb{R}^{d\times(d\cdot C)}$ compress the concatenated multi-channel features into $\mathbb{R}^d$. The matrix $\mathbf{W}_{\text{att}} \in \mathbb{R}^{C\times 2d}$ then maps the fused features to $C$ channel-specific scores. Subsequently, we obtain the normalized importance weights $\boldsymbol{w}_{e\to v}^{(l)} \in \mathbb{R}^C$ via a channel-wise Softmax over the neighborhood $\mathcal{E}_v$:

$$w_{e\to v,c}^{(l)} = \frac{\exp(s_{v,e,c}^{(l)}/\tau_a)}{\sum_{e'\in\mathcal{E}_v}\exp(s_{v,e',c}^{(l)}/\tau_a)}, \qquad (7)$$

where $\tau_a$ is the attention temperature coefficient.

*Remark.* We also explored explicit regularization strategies to enforce channel orthogonality. However, empirical results (detailed in Appendix B) indicate that MC-HNN naturally captures sufficient diversity, rendering additional constraints unnecessary.

**Dynamic Channel Gating and Update.** Simply increasing the number of channels may introduce redundancy or noise. To ensure that the expanded rank effectively contributes to the representation, we introduce a dynamic gating mechanism. A gate vector $\mathbf{g}_v^{(l)} \in \mathbb{R}^C$ is generated from the node's features to adaptively modulate the activation of each channel, similar to gating mechanisms employed in sequential models (Hochreiter & Schmidhuber, 1997; Cho et al., 2014) and GNNs (Bresson & Laurent, 2018):

$$\mathbf{g}_v^{(l)} = \text{Sigmoid}(\mathbf{W}_G\mathbf{x}_v^{(l-1)}), \qquad (8)$$

where $\mathbf{W}_G \in \mathbb{R}^{C\times(d\cdot C)}$. The final aggregated message $\mathbf{m}_v^{(l)}$ is obtained by weighting the incoming hyperedge messages and modulating them with the channel gate:

$$m_{v,c}^{(l)} = g_{v,c}^{(l)} \cdot \sum_{e\in\mathcal{E}_v}\left(w_{e\to v,c}^{(l)} \cdot h_{e,c}^{(l)}\right). \qquad (9)$$

Finally, the node representation for layer $l$ is updated by adding the aggregated message to the initial node projection $\mathbf{x}_v^{(0)}$:

$$\mathbf{x}_v^{(l)} = \text{ReLU}\left(\text{Norm}\left(\mathbf{x}_v^{(0)} + \mathbf{m}_v^{(l)}\right)\right). \qquad (10)$$

In our implementation, Norm$(\cdot)$ corresponds to Layer Normalization.

The time complexity of MC-HNN is analyzed in Appendix E. Our method maintains the same order of magnitude in efficiency as HGNN (Feng et al., 2019).

## 5. Discussions

In this section, we provide a theoretical discussion on how the proposed MC-HNN structurally addresses the limitations identified in Section 3.

### 5.1. Rank Capacity Analysis

Recall Proposition 3.2, which established that standard aggregation functions enforce a strict bottleneck of $\text{rank}(\mathbf{h}_e) \le 1$. Here, we discuss how MC-HNN expands this capacity boundary.

**Proposition 5.1** (Rank Upper Bound Expansion). *Let* $\mathbf{H}_e \in \mathbb{R}^{d \times C}$ *be the hyperedge representation matrix in MC-HNN, constructed by stacking $C$ channel-specific vectors. In a standard single-channel HNN, the rank is strictly bounded by 1. In contrast, for MC-HNN, assuming the projection weights $\{\mathbf{W}_{in}^{(c)}\}_{c=1}^{C}$ induce linearly independent subspaces, the upper bound of the representation capacity expands to:*

$$\mathrm{rank}(\mathbf{H}_e) \le \min(d, C). \tag{11}$$

*Remark.* MC-HNN transforms the aggregation target from a strictly constrained vector space (Rank-1) to a higher-capacity matrix manifold. This implies that regardless of the neighborhood size $|\mathcal{V}_e|$, the representation has a larger rank capacity, with an upper bound of $\min(d, C)$ instead of 1.

Detailed proof is provided in Appendix A.

### 5.2. Semantic Space Expansion

We next discuss the *Hyperedge Semantic Dependency*. We show that MC-HNN structurally breaks the rigid functional dependency on node features via latent encodings.

**Proposition 5.2** (Dependency Alleviation). *Standard aggregation restricts the hyperedge representation to be a deterministic function of its incident nodes, denoted as $\mathbf{h}_e = \phi(\mathbf{X}_{\mathcal{V}_e})$. Consequently, node aggregation alone may be insufficient to represent latent hyperedge factors $\mathbf{z}_e$ that contain information not predictable from the incident node features $\mathbf{X}_{\mathcal{V}_e}$. In contrast, MC-HNN introduces a learnable latent encoding $\tilde{\mathbf{t}}_e$, expanding the hypothesis space. This injection of additional hyperedge-level variables provides extra degrees of freedom to model such factors.*

*Remark.* This analysis motivates latent type encoding as an additional source of hyperedge-level flexibility, rather than treating hyperedges purely as passive aggregation results. It transforms the hyperedge from a passive relay into an active semantic entity, allowing the model to approximate complex relations where the whole is not merely the sum of its parts. Detailed proof is provided in Appendix A.

## 6. Experiments

### 6.1. Setup

**Datasets.** We evaluate MC-HNN on eight widely used real-world hypergraph benchmarks covering diverse domains, including co-citation networks Cora, CiteSeer, and PubMed (Yadati et al., 2019); the co-authorship network Cora-CA (Yadati et al., 2019); visual object classification datasets ModelNet40 (Wu et al., 2015) and NTU2012 (Chen et al., 2003); and heterophilic political networks Senate (Fowler, 2006) and House (Chodrow et al., 2021). The detailed

*Table 1.* Statistics of the experimental datasets.

| Dataset | Nodes | Edges | Classes | Features |
|---|---|---|---|---|
| Cora | 2,708 | 1,579 | 7 | 1,433 |
| CiteSeer | 3,312 | 1,709 | 6 | 3,703 |
| PubMed | 19,717 | 7,963 | 3 | 500 |
| Cora-CA | 2,708 | 1,072 | 7 | 1,433 |
| Senate | 282 | 315 | 2 | 100 |
| House | 1,290 | 340 | 2 | 100 |
| NTU2012 | 2,012 | 2,012 | 67 | 100 |
| ModelNet40 | 12,311 | 12,311 | 40 | 100 |

dataset statistics are summarized in Table 1. Note that for the featureless political networks Senate and House, we generate node features following the standard protocol established by Chien et al. (2022).

**Baselines.** We compare our proposed framework against state-of-the-art hypergraph neural networks, including HGNN (Feng et al., 2019), HCHA (Bai et al., 2021), HNHN (Dong et al., 2020), HyperGCN (Yadati et al., 2019), UniGC-NII (Huang & Yang, 2021), AllDeepSets (Chien et al., 2022), AllSetTransformer (Chien et al., 2022), HSDN (Hu et al., 2022), ED-HNN (Wang et al., 2023), FrameHGNN (Li et al., 2025), and Natural-HNN (Lee et al., 2026).

**Implementation Details.** To ensure a fair comparison, we strictly follow the experimental setting of Li et al. (2025). Each dataset is randomly divided into 50% training, 25% validation, and 25% testing sets. We report the average performance over 10 independent runs with random splits. Comprehensive hyperparameter configurations and implementation details are provided in Appendix F. Baseline results are directly taken from Li et al. (2025) where available. For HSDN and Natural-HNN, results on Cora, CiteSeer, PubMed, Cora-CA, NTU2012, and ModelNet40 are taken from Natural-HNN (Lee et al., 2026), while results on Senate and House are obtained under the same data split protocol with hyperparameters selected on the validation set.

### 6.2. Node Classification

The comparative results on node classification are summarized in Table 2. MC-HNN achieves competitive performance across datasets with varying scales, domains, and structural properties.

The most significant performance gains are observed on Senate and House, which are featureless and heterophilic networks. On these challenging benchmarks, standard HNNs (e.g., HGNN, UniGCNII) often struggle due to the absence of explicit node attributes and the complex, non-homophilous connectivity. Compared with FrameHGNN, MC-HNN improves accuracy by 10.7% on Senate and 4.7%

*Table 2.* Node classification accuracy on eight datasets. The best results are highlighted in **bold**.

| Method | Cora | CiteSeer | PubMed | Cora-CA | Senate | House | NTU2012 | ModelNet40 |
|---|---|---|---|---|---|---|---|---|
| HGNN | $79.39 \pm 1.36$ | $72.45 \pm 1.16$ | $86.44 \pm 0.44$ | $82.64 \pm 1.65$ | $48.59 \pm 4.52$ | $61.39 \pm 2.96$ | $87.72 \pm 1.35$ | $95.44 \pm 0.33$ |
| HCHA | $79.14 \pm 1.02$ | $72.42 \pm 1.42$ | $86.41 \pm 0.36$ | $82.55 \pm 0.97$ | $48.62 \pm 4.41$ | $61.36 \pm 2.53$ | $87.48 \pm 1.87$ | $94.48 \pm 0.28$ |
| HNHN | $76.36 \pm 1.92$ | $72.64 \pm 1.57$ | $86.90 \pm 0.30$ | $77.19 \pm 1.49$ | $50.93 \pm 6.33$ | $67.80 \pm 2.59$ | $89.11 \pm 1.44$ | $97.84 \pm 0.25$ |
| HyperGCN | $78.45 \pm 1.26$ | $71.28 \pm 0.82$ | $82.84 \pm 8.67$ | $79.48 \pm 2.08$ | $42.45 \pm 3.67$ | $48.32 \pm 2.93$ | $56.36 \pm 4.86$ | $75.89 \pm 5.26$ |
| UniGCNII | $78.81 \pm 1.05$ | $73.05 \pm 2.21$ | $88.25 \pm 0.40$ | $83.60 \pm 1.14$ | $49.30 \pm 4.25$ | $67.25 \pm 2.57$ | $89.30 \pm 1.33$ | $98.07 \pm 0.23$ |
| AllDeepSets | $76.88 \pm 1.80$ | $70.83 \pm 1.63$ | $88.75 \pm 0.33$ | $81.97 \pm 1.50$ | $48.17 \pm 5.67$ | $67.82 \pm 2.40$ | $88.09 \pm 1.52$ | $96.98 \pm 0.26$ |
| AllSetTransformer | $78.58 \pm 1.47$ | $73.08 \pm 1.20$ | $88.72 \pm 0.37$ | $83.63 \pm 1.47$ | $51.83 \pm 5.22$ | $69.33 \pm 2.20$ | $88.69 \pm 1.24$ | $98.20 \pm 0.20$ |
| HSDN | $76.63 \pm 1.51$ | $71.82 \pm 1.78$ | $87.19 \pm 0.32$ | $81.60 \pm 1.01$ | $67.57 \pm 4.74$ | $74.50 \pm 1.65$ | $89.72 \pm 1.20$ | $83.44 \pm 1.20$ |
| ED-HNN | $80.31 \pm 1.35$ | $73.70 \pm 1.38$ | $\mathbf{89.03 \pm 0.53}$ | $83.97 \pm 1.55$ | $64.79 \pm 5.14$ | $72.45 \pm 2.28$ | $88.67 \pm 0.92$ | $97.83 \pm 0.33$ |
| FrameHGNN | $81.51 \pm 0.99$ | $\mathbf{74.72 \pm 2.10}$ | $88.73 \pm 0.42$ | $85.18 \pm 0.69$ | $67.61 \pm 5.27$ | $72.82 \pm 2.22$ | $89.98 \pm 2.02$ | $98.41 \pm 0.18$ |
| Natural-HNN | $80.71 \pm 1.64$ | $73.29 \pm 1.74$ | $87.14 \pm 0.45$ | $84.99 \pm 0.49$ | $77.86 \pm 3.90$ | $77.33 \pm 1.73$ | $89.90 \pm 1.02$ | $\mathbf{98.56 \pm 0.30}$ |
| **MC-HNN** | $\mathbf{82.33 \pm 1.65}$ | $74.48 \pm 1.29$ | $88.93 \pm 0.46$ | $\mathbf{86.32 \pm 1.30}$ | $\mathbf{78.31 \pm 3.20}$ | $\mathbf{77.59 \pm 2.72}$ | $\mathbf{90.82 \pm 1.09}$ | $98.52 \pm 0.19$ |

on House. Moreover, MC-HNN remains competitive with recent disentangled hypergraph baselines, slightly outperforming Natural-HNN on both Senate and House in mean accuracy. This empirical evidence is consistent with our motivation that node-independent hyperedge information can be useful in featureless and heterophilic settings.

On standard co-citation (Cora, PubMed) and co-authorship (Cora-CA) networks, MC-HNN maintains top-tier performance. It achieves the highest accuracy on Cora and Cora-CA. Regarding PubMed and CiteSeer, although MC-HNN does not rank first, the performance gap is marginal ($< 0.3\%$) and falls well within the standard deviation ranges. This indicates that our model performs statistically on par with leading methods like ED-HNN and FrameHGNN. We attribute this saturation to the relatively simple homophilic structures of these datasets, where advanced semantic decoupling provides diminishing returns compared to standard smoothing. Notably, on the visual object classification datasets, MC-HNN achieves the best performance on NTU2012 and remains highly competitive on ModelNet40, demonstrating its robust generalization capability in handling dense geometric structures.

Because standard hypergraph benchmarks may not contain sufficiently rich or interpretable hyperedge-level semantics, we further evaluate MC-HNN on cancer subtype classification datasets with biologically grounded high-order relations. In these datasets, each hyperedge corresponds to a biological pathway, providing functional contexts that are potentially informative for disease subtype prediction. The full experimental protocol and results are reported in Appendix C. MC-HNN achieves competitive performance across six TCGA cancer datasets, obtaining the best results on four datasets and the second-best results on the remaining two.

### 6.3. Over-smoothing Analysis

To evaluate the robustness of MC-HNN, we conduct experiments with model depths ranging from 2 to 64 layers on

Cora, House, and Cora-CA. We compared our method with HGNN, UniGCNII, and ED-HNN. As shown in Figure 3, HGNN exhibits drastic performance deterioration as depth exceeds 4 layers, confirming the severity of feature collapse in standard spectral aggregation. While ED-HNN shows moderate resilience, it eventually degrades significantly at extreme depths (32/64 layers). Similar to UniGCNII, MC-HNN effectively maintains capability in deep networks; however, it offers superior representation quality and overall performance.

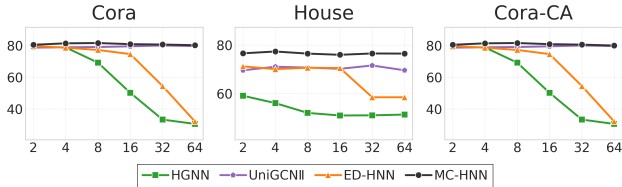

*Figure 3.* Over-smoothing analysis on different datasets. The X-axis represents the model depth (number of layers), and the Y-axis denotes the node classification accuracy (%).

We attribute this inherent robustness to two core theoretical designs in our framework. First, to mitigate rank collapse, the multi-channel projection mechanism maintains high-rank representations by enforcing feature diversity across different subspaces, effectively preventing high-frequency information from becoming excessively mixed during recursive propagation. Second, the latent hyperedge type encoding can provide additional hyperedge-level flexibility, helping the model maintain more diverse representations during deep propagation.

### 6.4. Ablation Study

To investigate the contribution of each component in MC-HNN, we conduct ablation studies on three representative datasets (Cora, Senate, and NTU2012) by removing the following modules: (1) **w/o multi-channel**, which replaces the multi-channel mechanism with a single-channel variant, where the hidden dimension is scaled to ensure the total

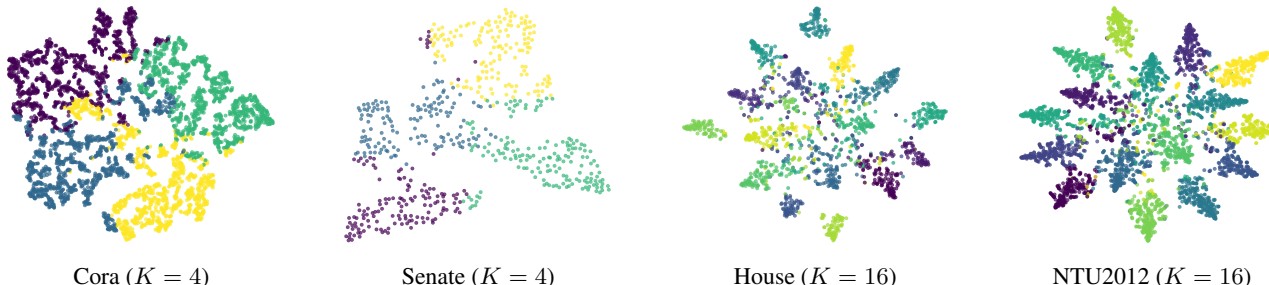

Cora ($K = 4$)  Senate ($K = 4$)  House ($K = 16$)  NTU2012 ($K = 16$)

*Figure 4.* t-SNE visualization of learned latent hyperedge type encodings ($\tilde{\mathbf{t}}_e$) on Cora, Senate, House, and NTU2012.

*Table 3.* Ablation study on component effectiveness.

| Method | Cora | Senate | NTU2012 |
|---|---|---|---|
| MC-HNN (Full) | $82.33 \pm 1.65$ | $78.31 \pm 3.20$ | $90.82 \pm 1.09$ |
| w/o multi-channel | $78.73 \pm 1.62$ | $77.32 \pm 4.81$ | $90.36 \pm 1.29$ |
| w/o latent types | $81.64 \pm 1.59$ | $76.62 \pm 2.59$ | $90.70 \pm 0.49$ |
| w/o gating | $82.02 \pm 1.01$ | $77.61 \pm 4.22$ | $90.46 \pm 1.18$ |

number of parameters remains consistent; (2) **w/o latent types**, which removes the latent hyperedge type encoding $\tilde{\mathbf{t}}_e$; and (3) **w/o gating**, which removes the adaptive gating mechanism. The results are reported in Table 3.

Removing the multi-channel mechanism causes the most significant performance drop, particularly on the Cora dataset (a decline of 3.6%). This dataset contains rich textual node features, and this result aligns with our analysis that mapping high-dimensional features into a single low-dimensional space exacerbates rank collapse. By utilizing multiple subspaces, the full model effectively preserves high-frequency information and maintains the expressive power of node representations.

The removal of latent types leads to degradation across all datasets, with the most notable drop observed on Senate. This confirms that the learned $\tilde{\mathbf{t}}_e$ provides useful hyperedge-level degrees of freedom that are not available from node aggregation alone. Especially in datasets like Senate where structural patterns are dominant, the latent types act as a crucial additional hyperedge-level signal enabling the model to distinguish between functionally different hyperedges even when node feature signals are ambiguous.

Finally, removing the adaptive gating (w/o gating) results in a consistent, albeit milder, performance decline across all three datasets. This suggests that while the multi-channel structure and latent types are the primary drivers of performance, the adaptive gating mechanism plays a vital role in fine-tuning the aggregation. It successfully identifies the reliability of different channels, allowing the model to dynamically balance the information flow and filter out noise for stable optimization.

For NTU2012, the performance drops are less pronounced. This is likely because its node features are high-level visual representations extracted by deep networks, which are inherently discriminative. These strong features tend to dominate the classification, leaving limited room for improvement via structural refinement.

### 6.5. Visualization of Latent Hyperedge Types

To inspect the learned latent hyperedge type encodings, we visualize the learned latent codes $\tilde{\mathbf{t}}_e$ using t-SNE (Maaten & Hinton, 2008) on four representative datasets: Cora, Senate, House, and NTU2012. The optimal codebook size $K$ for each dataset was selected via a grid search over $\{2, 4, 8, 12, 16\}$ based on validation performance. Specifically, the model converges to $K = 4$ for Cora and Senate, and $K = 16$ for House and NTU2012.

As illustrated in Figure 4, the learned encodings organize into well-separated clusters across all datasets under the selected codebook sizes.

A critical insight emerges when comparing the learned hyperedge types with the ground-truth node labels. In the political networks (Senate and House), although nodes belong to only two classes (i.e., parties), the model identifies a richer set of structural types ($K = 4$ and 16, respectively). This suggests that the learned codes may capture variation beyond the binary node labels, although we do not claim that these clusters directly correspond to known semantic categories.

Moreover, the variation in optimal $K$ underscores a dependency on the intrinsic structural complexity. For instance, while both political datasets share the same label space, the significantly larger House network (1,290 nodes) necessitates a richer codebook ($K = 16$) to encapsulate its broader range of association patterns compared to the smaller Senate network. Similarly, for NTU2012, the model saturates the search space at $K = 16$, indicating a strong demand for high-dimensional semantic capacity to match the complex geometric dependencies in object classification.

# 7. Conclusion

In this paper, we revisited the prevalent unified paradigm of Hypergraph Neural Networks and analyzed two structural bottlenecks: rank collapse and hyperedge semantic dependency. To address these challenges, we proposed the Multi-Channel Hypergraph Neural Network (MC-HNN). By establishing a multi-channel message passing mechanism, our framework effectively expands the representation capacity, preventing feature indistinguishability in deep layers. Furthermore, the introduction of the latent hyperedge type encoding introduces additional hyperedge-level degrees of freedom beyond aggregated node attributes. Empirical results show that MC-HNN achieves competitive performance across diverse benchmarks.

# Acknowledgements

We want to thank all the anonymous reviewers for their constructive comments. This work was financially supported by the National Natural Science Foundation of China (Grant No. 62272399), the Science and Technology Project of the Ministry of Industry and Information Technology (Grant No. CEIEC-2025-ZM02-0014-21), and the Fundamental Research Funds for the Central Universities (Grant No. 20720250004).

# Impact Statement

This paper presents work whose goal is to advance the field of machine learning, particularly representation learning on hypergraphs. Our experiments include public benchmark datasets from citation, visual, political, and biomedical domains. Although some experiments involve political networks and cancer genomics benchmarks, our work is methodological in nature and is not intended to make claims about individual political actors, nor to support clinical diagnosis, treatment recommendation, or direct deployment in healthcare decision-making systems. The biomedical datasets are processed following established public benchmark protocols, and the results should be interpreted as benchmark evaluations rather than validated clinical evidence. There are many potential societal consequences of machine learning research, but beyond the considerations above, none of them need to be specifically highlighted here.

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

# A. Proofs

## A.1. Proof of Proposition 3.2

*Proof.* Consider the local feature matrix $\mathbf{X}_{\mathcal{V}_e} \in \mathbb{R}^{F \times k}$. Let $r = \mathrm{rank}(\mathbf{X}_{\mathcal{V}_e})$. Under the assumption of semantic diversity among nodes and high-dimensional feature space ($F \gg k$), the column vectors are typically linearly independent, implying $r = k$ (full column rank). Even in cases of partial redundancy, we assume $r > 1$.

The aggregation operation $\mathcal{A}_{\mathcal{V} \to \mathcal{E}}$ maps the set of vectors $\{\mathbf{x}_u\}_{u \in \mathcal{V}_e}$ to a single hyperedge representation vector $\mathbf{h}_e \in \mathbb{R}^F$. By the fundamental definition of matrix rank, the rank of a single vector $\mathbf{h}_e$ (which can be viewed as an $F \times 1$ matrix) is strictly bounded by 1 (specifically, 1 if non-zero, 0 otherwise).

Consequently, the aggregation forces a projection from a rank-$r$ subspace to a rank-1 subspace. The dimension of the linear subspace spanned by the representation collapses from $r$ to at most 1. The minimal loss in degrees of freedom is therefore:

$$\Delta = \mathrm{rank}(\mathbf{X}_{\mathcal{V}_e}) - \mathrm{rank}(\mathbf{h}_e) \geq r - 1. \tag{12}$$

Since $r > 1$, this compression reduces the rank capacity of the intermediate hyperedge representation, thereby proving the proposition. $\square$

## A.2. Proof of Proposition 5.1

*Proof.* Let $\mathbf{H}_e = [\mathbf{h}_{e,1}^{(l)}, \ldots, \mathbf{h}_{e,C}^{(l)}] \in \mathbb{R}^{d \times C}$ denote the hyperedge representation matrix formed by stacking outputs from $C$ distinct channels, where $d$ is the feature dimension per channel.

The rank of any matrix is universally bounded by its dimensions (the minimum of its number of rows and columns).

First, considering the column space, $\mathbf{H}_e$ consists of $C$ column vectors. Thus, $\mathrm{rank}(\mathbf{H}_e) \leq C$. This reflects the diversity introduced by the multi-channel mechanism.

Second, considering the row space, the matrix lies in a $d$-dimensional feature space. Thus, its rank is naturally constrained by the feature dimension: $\mathrm{rank}(\mathbf{H}_e) \leq d$.

Combining these algebraic constraints yields the correct upper bound:

$$\mathrm{rank}(\mathbf{H}_e) \leq \min(d, C). \tag{13}$$

In a standard single-channel HNN, the representation is a single vector ($C = 1$), enforcing $\mathrm{rank}(\mathbf{h}_e) \leq 1$. In MC-HNN, provided that $d \geq C$ (which is typical in high-dimensional embeddings) and the channel projections are linearly independent, the rank upper bound increases from 1 to at most $C$, expanding the rank capacity of the hyperedge representation. $\square$

## A.3. Proof of Proposition 5.2

*Proof.* Consider the standard aggregation paradigm where the hyperedge representation is derived from node features: $\mathbf{h}_e = \mathcal{A}(\{\mathbf{x}_v\}_{v \in \mathcal{V}_e})$. Regardless of the complexity of $\mathcal{A}$, the resulting representation remains a deterministic function of the incident node features $\mathbf{X}_{\mathcal{V}_e}$.

Suppose a downstream-relevant hyperedge factor $\mathbf{z}_e$ depends on both the incident node features and an additional hyperedge-level variable $\mathbf{u}_e$, i.e., $\mathbf{z}_e = g(\mathbf{X}_{\mathcal{V}_e}, \mathbf{u}_e)$. When $\mathbf{u}_e$ contains information not predictable from $\mathbf{X}_{\mathcal{V}_e}$, a representation that is only a function of $\mathbf{X}_{\mathcal{V}_e}$ may be insufficient to encode this factor.

MC-HNN introduces a learnable latent encoding $\tilde{\mathbf{t}}_e$ for each hyperedge, yielding a representation of the form $\mathbf{h}_e' = \mathcal{A}'(\mathbf{X}_{\mathcal{V}_e}, \tilde{\mathbf{t}}_e)$. This expands the hypothesis space by adding hyperedge-level degrees of freedom beyond aggregated node features, which can help model such factors. $\square$

# B. Investigation on Explicit Channel Diversity Constraints

## B.1. Motivation and Formulation

A natural question regarding multi-channel architectures is whether the distinct channels learn diverse representations or degenerate into redundant subspaces. To investigate this, during the development of MC-HNN, we explored two explicit regularization strategies designed to force channel decoupling and attention balance.

**1. Channel Decoupling Regularization ($\mathcal{L}_{\textbf{decouple}}$).** To encourage channels to learn disparate features, we formulated a regularization term based on the orthogonality of the channel representations. Let $\mathbf{C} \in \mathbb{R}^{C \times d}$ denote the matrix of channel representations for a node (after normalization). We aimed to minimize the correlation between different channels:

$$\mathcal{L}_{\text{decouple}} = \|\mathbf{C}\mathbf{C}^T - \mathbf{I}\|_F^2, \tag{14}$$

where $\mathbf{I}$ is the identity matrix and $\|\cdot\|_F$ denotes the Frobenius norm. This term penalizes non-orthogonal channel vectors, theoretically forcing them to capture independent semantic directions.

**2. Attention Entropy Regularization ($\mathcal{L}_{\textbf{balance}}$).** To prevent specific channels from dominating the attention weights (Attention Collapse), we proposed an entropy-based regularizer. We incorporated a hyperedge-size-dependent weight $\log(|\mathcal{V}_e|)$ to encourage larger hyperedges—which contain richer information—to utilize a more balanced attention distribution across channels:

$$\mathcal{L}_{\text{balance}} = -\frac{1}{|\mathcal{J}|} \sum_{(v,e) \in \mathcal{J}} \mathcal{H}(\boldsymbol{\alpha}_{v,e}) \cdot \log(|\mathcal{V}_e|), \tag{15}$$

where $\mathcal{H}(\cdot)$ is the Shannon entropy of the attention weights, and $\mathcal{J}$ is the set of node-hyperedge incidences. The term $\log(|\mathcal{V}_e|)$ acts as a scaling factor: for small hyperedges, we allow focused attention (low entropy); for large hyperedges, we encourage exploration of internal information (high entropy).

The total training loss was formulated as: $\mathcal{L} = \mathcal{L}_{\text{main}} + \lambda_1 \mathcal{L}_{\text{decouple}} + \lambda_2 \mathcal{L}_{\text{balance}}$.

### B.2. Empirical Results and Analysis

We compared the performance of the standard MC-HNN (without explicit regularization) against the regularized version. To ensure a rigorous evaluation, we conducted a comprehensive grid search for the hyperparameters $\lambda_1$ and $\lambda_2$ over the search space $\{1e\text{-}4, 1e\text{-}3, 5e\text{-}3, 1e\text{-}2, 5e\text{-}2, 1e\text{-}1, 5e\text{-}1, 0.8\}$. The results reported for MC-HNN (+Reg) in Table 4 correspond to the best configuration identified from this extensive search.

*Table 4.* Performance comparison between MC-HNN with and without explicit diversity regularization. The best results are highlighted in **bold**.

| Method | Cora | CiteSeer | PubMed | Cora-CA | Senate | House | NTU2012 | ModelNet40 |
|---|---|---|---|---|---|---|---|---|
| MC-HNN (Standard) | **82.33 ± 1.65** | **74.48 ± 1.29** | 88.93 ± 0.46 | **86.32 ± 1.30** | **78.31 ± 3.20** | 77.59 ± 2.72 | **90.82 ± 1.09** | 98.52 ± 0.19 |
| MC-HNN (+Reg) | 82.04 ± 1.67 | 73.99 ± 1.45 | **88.98 ± 0.32** | 85.88 ± 1.17 | 77.89 ± 2.97 | **78.39 ± 2.62** | 90.82 ± 1.02 | **98.53 ± 0.19** |

**Discussion.** As evidenced in Table 4, even with exhaustive hyperparameter tuning, the introduction of explicit regularization does not yield a consistent performance advantage. Specifically, on homophilic benchmarks such as Cora, CiteSeer, and Cora-CA, the standard MC-HNN consistently outperforms the regularized variant. This suggests that the strict orthogonality constraint ($\mathcal{L}_{\text{decouple}}$) imposes an excessive rigidity, forcing the model to discard naturally correlated semantic features that are beneficial for inference. Conversely, while the regularized model achieves marginal gains on datasets like PubMed and House (e.g., $< 0.1\%$ improvement), these slight increments fail to justify the increased computational complexity and the need for sensitive hyperparameter tuning. Adhering to the principle of Occam's Razor, we prioritize the simpler and more robust standard design, as "mild" natural diversity proves sufficient for high-performance representation learning.

## C. Cancer Subtype Classification on Biomedical Hypergraphs

We further evaluate MC-HNN on cancer subtype classification datasets with richer and more interpretable hyperedge-level semantics. Following Natural-HNN (Lee et al., 2026), we use six TCGA cancer subtype classification datasets whose preprocessing follows Pathformer (Liu et al., 2024). Each patient is represented as a hypergraph, where nodes correspond to genes and hyperedges correspond to biological pathways. Thus, unlike generic benchmark hypergraphs whose hyperedges may not carry clear semantic factors, these biomedical hyperedges encode pathway-level functional relations among genes. The node features are derived from multi-omics measurements from TCGA (Weinstein et al., 2013), and the graph-level task is to predict the cancer subtype of each patient. We evaluate on six datasets, including BRCA, STAD, SARC, LGG, HNSC, and CESC. This setting directly tests whether MC-HNN can exploit latent hyperedge-level semantic factors that are potentially informative for disease subtype prediction.

We follow the evaluation protocol of Natural-HNN. For each dataset, we use class-balanced 50%/25%/25% train/validation/test splits over 10 runs and report Macro-F1. Baseline results are taken from Natural-HNN. As shown in Table 5, MC-HNN achieves the best performance on four out of six datasets and remains close to the best-performing method on the other two, with differences falling within the reported standard deviations. These results indicate that MC-HNN remains competitive in biomedical hypergraph classification and can be applied beyond standard node classification benchmarks to settings with richer domain-specific hyperedge semantics.

*Table 5.* Cancer subtype classification results in Macro-F1. Baseline results are from Natural-HNN (Lee et al., 2026); ∗ denotes the multi-head attention variant. The best result is highlighted in **bold**.

| Method | BRCA | STAD | SARC | LGG | HNSC | CESC |
|---|---|---|---|---|---|---|
| HGNN | $0.726 \pm 0.053$ | $0.563 \pm 0.040$ | $0.684 \pm 0.067$ | $0.694 \pm 0.033$ | $0.799 \pm 0.053$ | $0.835 \pm 0.052$ |
| HCHA | $0.704 \pm 0.051$ | $0.558 \pm 0.044$ | $0.675 \pm 0.068$ | $0.682 \pm 0.041$ | $0.783 \pm 0.055$ | $0.844 \pm 0.054$ |
| HNHN | $0.697 \pm 0.046$ | $0.573 \pm 0.072$ | $0.688 \pm 0.075$ | $0.674 \pm 0.038$ | $0.791 \pm 0.035$ | $0.837 \pm 0.059$ |
| UniGCNII | $0.697 \pm 0.052$ | $0.617 \pm 0.059$ | $0.728 \pm 0.066$ | $0.663 \pm 0.039$ | $0.830 \pm 0.030$ | $0.841 \pm 0.046$ |
| AllDeepSets | $0.716 \pm 0.058$ | $0.557 \pm 0.044$ | $0.599 \pm 0.058$ | $0.665 \pm 0.046$ | $0.801 \pm 0.058$ | $0.870 \pm 0.044$ |
| AllSetTransformer | $0.743 \pm 0.057$ | $0.553 \pm 0.046$ | $0.719 \pm 0.052$ | $0.653 \pm 0.038$ | $0.814 \pm 0.036$ | $0.847 \pm 0.046$ |
| HyperGAT | $0.637 \pm 0.121$ | $0.534 \pm 0.063$ | $0.574 \pm 0.153$ | $0.665 \pm 0.054$ | $0.789 \pm 0.061$ | $0.832 \pm 0.046$ |
| HyperGAT∗ | $0.641 \pm 0.115$ | $0.502 \pm 0.087$ | $0.584 \pm 0.150$ | $0.646 \pm 0.043$ | $0.791 \pm 0.079$ | $0.827 \pm 0.041$ |
| SHINE | $0.446 \pm 0.155$ | $0.371 \pm 0.135$ | $0.529 \pm 0.160$ | $0.628 \pm 0.104$ | $0.718 \pm 0.055$ | $0.745 \pm 0.159$ |
| SHINE∗ | $0.651 \pm 0.053$ | $0.532 \pm 0.064$ | $0.673 \pm 0.059$ | $0.650 \pm 0.046$ | $0.770 \pm 0.040$ | $0.837 \pm 0.061$ |
| HSDN | $0.757 \pm 0.044$ | $0.629 \pm 0.045$ | $0.726 \pm 0.063$ | $0.692 \pm 0.038$ | $0.811 \pm 0.044$ | $0.867 \pm 0.033$ |
| ED-HNN | $0.735 \pm 0.047$ | $0.615 \pm 0.050$ | $0.718 \pm 0.071$ | $0.700 \pm 0.030$ | $0.835 \pm 0.047$ | $0.875 \pm 0.053$ |
| ED-HNNII | $0.722 \pm 0.045$ | $0.536 \pm 0.057$ | $0.650 \pm 0.087$ | $0.695 \pm 0.039$ | $0.845 \pm 0.025$ | $\mathbf{0.895 \pm 0.044}$ |
| Natural-HNN | $0.804 \pm 0.036$ | $\mathbf{0.659 \pm 0.049}$ | $0.745 \pm 0.045$ | $0.707 \pm 0.035$ | $0.862 \pm 0.045$ | $0.881 \pm 0.042$ |
| MC-HNN | $\mathbf{0.809 \pm 0.029}$ | $0.651 \pm 0.030$ | $\mathbf{0.760 \pm 0.067}$ | $\mathbf{0.711 \pm 0.032}$ | $\mathbf{0.872 \pm 0.023}$ | $0.888 \pm 0.044$ |

# D. Hyperparameter Sensitivity Analysis

To investigate the stability of MC-HNN and the impact of its two core hyperparameters, namely the number of channels $C$ and the number of latent hyperedge types $K$, we conduct sensitivity experiments on two representative datasets: Cora (feature-rich, homophilic) and Senate (featureless, heterophilic). The results are illustrated in Figure 5.

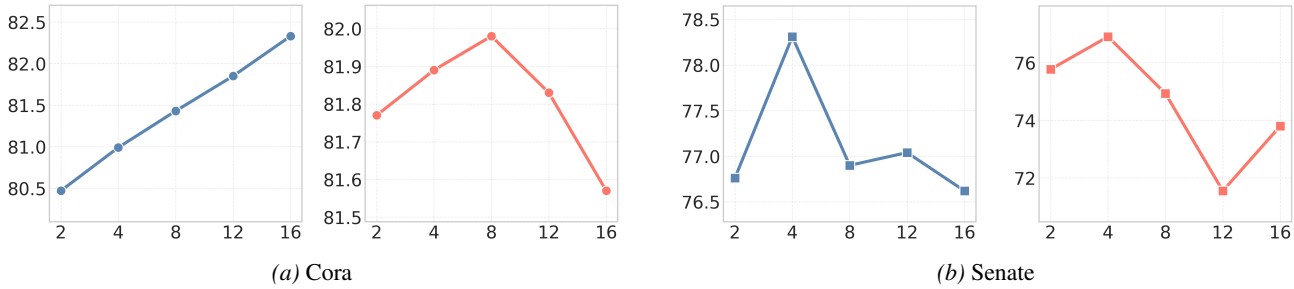

*(a)* Cora             *(b)* Senate

*Figure 5.* Hyperparameter sensitivity analysis on Cora and Senate. In each subfigure, the **left plot (Blue)** shows the impact of varying the number of channels $C$, while the **right plot (Red)** shows the impact of varying the number of latent types $K$. The Y-axis represents node classification accuracy (%), and the X-axis represents the hyperparameter values.

**Impact of Channel Count $C$.** The blue curves in Figure 5 illustrate how the number of channels affects performance. On the feature-rich Cora dataset, accuracy consistently improves as $C$ increases from 2 to 16. This monotonic growth empirically validates our theoretical claim in Proposition 5.1: increasing the number of channels effectively expands the rank upper bound ($\min(|\mathcal{V}_e|, C)$), allowing the model to capture more diverse signals from the neighborhood without information collapse. Conversely, on the smaller Senate dataset (282 nodes), performance peaks at $C = 4$ and then declines. This suggests that for small-scale graphs, an excessive number of channels may induce over-parameterization, leading to overfitting. Thus, the optimal $C$ serves as a trade-off between representation capacity and model complexity relative to data

scale.

**Impact of Latent Types $K$.** The red curves depict the sensitivity to the codebook size $K$. Both datasets exhibit a bell-shaped trend, indicating the existence of an optimal "semantic granularity." On Cora, the performance peaks at $K = 8$. A codebook that is too small $K = 2$ fails to capture the sufficient diversity of hyperedge functions, while an overly large codebook ($K = 16$) may introduce redundancy or noise, fragmenting the semantic clusters. Similarly, Senate achieves its best performance at $K = 4$.

## E. Time Complexity

We explicitly analyze the computational complexity of the proposed MC-HNN layer. Let $N$ and $M$ denote the number of nodes and hyperedges, respectively, and let $\Omega = \|\mathbf{H}\|_0$ represent the number of non-zero entries in the incidence matrix.

The computational cost of our layer consists of three main components: linear projections, latent type generation, and sparse message aggregation. Projecting features from the total dimension $C \cdot d$ to the subspace dimension $d$ (and vice versa) for both nodes and hyperedges requires $O((N + M) \cdot C \cdot d^2)$. Generating latent type encodings via the global codebook involves a weighted combination of prototypes, costing $O(M \cdot K \cdot C \cdot d)$, where $K$ is the number of prototypes. Calculating attention scores and performing weighted aggregation involves traversing the incidence structure. This scales with the number of non-zero entries, costing $O(\Omega \cdot C \cdot d)$.

Summing these components, the total complexity per layer is:

$$O \left( \underbrace{(N + M) \cdot C \cdot d^2}_{\text{Projections}} + \underbrace{M \cdot K \cdot C \cdot d}_{\text{Type Gen.}} + \underbrace{\Omega \cdot C \cdot d}_{\text{Aggregation}} \right). \tag{16}$$

For comparison, a standard HGNN (Feng et al., 2019) with a hidden dimension of $d$ has a complexity of $O(N \cdot d^2 + \Omega \cdot d)$. Our proposed model scales linearly with the number of channels $C$. Furthermore, since the number of prototypes $K$ is typically a small constant (e.g., $K \ll d$), the additional cost for latent type generation is comparable to the aggregation cost and does not alter the overall complexity order.

## F. Experimental Setup

All experiments were implemented using PyTorch and conducted on a single NVIDIA RTX3090 GPU with 24GB of memory. To ensure a fair evaluation, we randomly split each dataset into training, validation, and testing sets with a ratio of 50%/25%/25%, respectively. We report the average performance and standard deviation over 10 independent runs with different random seeds. The model was optimized using the Adam optimizer. For hyperparameter tuning, we performed a grid search on the validation set to select the optimal configuration. The detailed hyperparameter search space is summarized in Table 6.

*Table 6.* Search space for hyperparameters.

| Hyperparameter | Range |
|---|---|
| Learning rate | $\{1e\text{-}4, 1e\text{-}3, 5e\text{-}3, 1e\text{-}2, 5e\text{-}2\}$ |
| Weight decay | $\{1e\text{-}5, 1e\text{-}4, 5e\text{-}4, 1e\text{-}3, 5e\text{-}3, 1e\text{-}2, 5e\text{-}2\}$ |
| Dropout | $\{0.2, 0.3, 0.4, 0.5, 0.6, 0.7, 0.8\}$ |
| Hidden dimension | $\{16, 32, 64, 128, 256\}$ |
| $\tau_s$ | $\{0.5, 1.0, 2.0\}$ |
| $\tau_a$ | $\{0.5, 1.0, 2.0\}$ |
| Number of layers | $\{1, 2, 3, 4, 5\}$ |
| Number of latent prototypes | $\{2, 4, 8, 12, 16\}$ |
| Number of channels | $\{2, 4, 8, 12, 16\}$ |

