# OpenReview forum: "MC-HNN: Learning Latent Structural Semantics and High-Rank Representations for Hypergraph Neural Networks"
_ICML.cc/2026/Conference — ICML 2026 regular_

### Official Review · Reviewer_GaNT · 2026-02-23

**Soundness:** 2
**Presentation:** 3
**Significance:** 2
**Originality:** 2
**Overall Recommendation:** 3
**Confidence:** 5

**Summary:**

The paper addresses a relevant problem in hypergraph neural networks with a clean, well-motivated framework. The architectural design is sensible, and the empirical results—particularly on heterophilic and featureless benchmarks—are encouraging. However, the theoretical contributions are significantly overstated relative to their actual depth, and several key empirical validations are missing (rank measurement, runtime, statistical tests, scalability). The novelty is moderate given existing multi-channel GNN literature.

**Compliance With Llm Reviewing Policy:**

Affirmed.

**Final Justification:**

Thanks for the authors' response. However, the paper is not outstanding enough for ICML. I hold my prior evaluations.

**Key Questions For Authors:**

1. The so-called rank collapse problem is confusing，why the phenomenon in Proposition 3.2 theoretically prohibiting the preservation of diverse signals from multiple nodes simultaneously? Can you provide empirical measurements of the effective rank (or singular value spectrum) of hyperedge representations in baseline HNNs versus MC-HNN? This would be the most direct validation of the rank collapse hypothesis and would significantly strengthen the theoretical narrative.

2. The current “rank collapse” and “semantic dependency” arguments are more suggestive than rigorous. Discussions parts read as algebraic observations rather than substantive bounds on learning capacity.

3. Do the learned latent hyperedge types correlate with any known structural or semantic properties of the hyperedges?

4. What exactly drives the big improvements on featureless political datasets? Is it mostly latent types (structural anchors), multi-channel capacity, or attention/gating? A targeted ablation on Senate/House that includes “ID embedding only” and “type encoding only” would clarify.

**Limitations:**

1. The novelty of multi-channel message passing for hypergraphs is less prominent.

2. The empirical validation of the rank collapse phenomenon is entirely absent—the paper never measures actual representation rank in practice.

**Strengths And Weaknesses:**

Strengtins:

1. The paper is generally well-written and logically organized.

2. The paper introduces a sensible and well-integrated architectural design.

3. The experimental evaluation is performed comprehensively, and strong empirical results are obtained on several challenging settings.

Weakness:

1. The theoretical contributions are overstated and somewhat trivial.

        a. The Proposition 3.2  is essentially a restatement of the fact that projecting a set of vectors onto a single vector reduces rank to 1. This is definitionally true and does not constitute a deep theoretical insight. The proposition does not quantify what information is lost or how much task-relevant information is destroyed—it merely states a dimensional bound.

        b. The Proposition 5.1 states that a d x C matrix has rank at most min(d, C), which is again a basic linear algebra fact. The critical assumption—that "projection weights induce linearly independent subspaces"—is non-trivial and is never formally verified or guaranteed by the training procedure.

2. Limited novelty of the multi-channel mechanism. While the theoretical framing via "Rank Collapse" is novel, the "Multi-Channel" mechanism has been well established in many recently proposed HGNN frameworks. For example, in MHCN[1], HCCF[2], Motifs-Res[3], and MHCPL[4] information Aggregation and message passing are performed in a multi-channel construction. The MC-HNN proposed in this paper bears a strong structural resemblance to these methods.The authors must sufficiently differentiate their approach from existing multi-channel HGNN frameworks and emphasize their unique contributions.

3. The relationship between rank collapse and actual performance degradation is not empirically validated.The paper claims rank collapse causes information loss and performance degradation, but never measures the actual rank of hyperedge representations in baseline models versus MC-HNN.

4. The latent hyperedge type encoding introduces a significant number of free parameters without clear regularization. And, there is no mechanism ensuring that these learned types correspond to meaningful semantic categories.

[1] Self-Supervised Multi-Channel Hypergraph Convolutional Network for Social Recommendation. In WWW, 2021.

[2] Hypergraph Contrastive Collaborative Filtering. In SIGIR, 2022.

[3] Motifs-based Recommender System via Hypergraph Convolution and Contrastive Learning. Neurocomputing, 2022, 512: 323–338.

[4] Multi-view Hypergraph Contrastive Policy Learning for Conversational Recommendation. In SIGIR, 2023.

---

> ### Author Rebuttal · Authors · 2026-03-31
>
> We thank Reviewer GaNT for the detailed review. Below we address the comments point by point with new evidence and clarifications.
>
> **Q1: Why does Proposition 3.2 lead to the failure of retaining diverse signals? Is there any empirical rank measurement?**
>
> Proposition 3.2 aims to illustrate a structural fact: in standard HNNs, each hyperedge is compressed into a single vector before being rebroadcast to the nodes, forcing diverse node signals through a rank-1 bottleneck. We now directly verify this on trained baseline HNNs:
>
> | Dataset | UniGCNII | ED-HNN | MC-HNN (per-HE effective rank) |
> | ------- | -------- | ------ | ------------------------------ |
> | Cora    | **1.00** | **1.00** | **9.51** |
> | Senate  | **1.00** | **1.00** | **9.64** |
> | House   | **1.00** | **1.00** | **1.94** |
>
> For trained UniGCNII and ED-HNN, the extracted hyperedge representations remain single vectors, so the rank of each hyperedge is exactly 1 by construction. In contrast, MC-HNN generates a $d \times C$ matrix for each hyperedge, and the table reports its empirical effective rank. Therefore, the baseline behavior is not merely theoretical: trained baseline HNNs remain at rank 1, whereas MC-HNN learns significantly richer hyperedge representations.
>
> **Q2: The current theoretical arguments act more like suggestive observations rather than rigorously justified bounds.**
>
> We agree and will tone down the claims in the paper accordingly. Propositions 3.2 and 5.1 are better suited as observations/analyses rather than deep, stand-alone theories. We will uniformly adjust the tone in the abstract, contribution list, and theoretical discussions, narrowing our core claim to: the paper formalizes a structural bottleneck and proposes an architecture that empirically and measurably mitigates it.
>
> **Q3: Are the learned latent types related to known structural or semantic properties?**
>
> The benchmarks used in this work do not provide explicit hyperedge-level semantic labels or hyperedge features, so there is no target for directly examining alignment with known semantic categories. The current evidence supports functional structural utility: First, the codebook-vs-direct comparison (see Q3 for Reviewer i6bT) shows that a structured shared codebook outperforms unconstrained per-hyperedge embeddings across 4 test datasets while using fewer parameters. Second, on the political datasets without node features, a targeted diagnostic on Senate (see Q4 for Reviewer GaNT below) shows that Types-only performs significantly closer to the full model than ID-only (74.65 ± 5.23 vs 70.00 ± 5.12), indicating that the gains cannot be simply reduced to hyperedge-ID memorization.
>
> **Q4: What exactly drives the significant improvements on Senate/House?**
>
> We have completed the targeted ablations requested by the reviewer. To keep the rebuttal numerically aligned with the paper, we use the submitted-version full-model result as the public reference point, and report the new Types-only / ID-only numbers as reviewer-requested diagnostics:
>
> | Dataset | Full MC-HNN (submitted) | Types-only | ID-only |
> |---------|--------------------------|------------|---------|
> | Senate | **78.31 ± 3.20** | 74.65 ± 5.23 | 70.00 ± 5.12 |
> | House | **77.59 ± 2.72** | 77.83 ± 1.83 | 76.41 ± 1.65 |
>
> On Senate, Types-only is much closer to the full model than ID-only, indicating that the gain is not reducible to hyperedge-ID memorization and that latent types are the main structural anchor. On House, the smaller gap suggests a more mixed source of gain.
>
> Additionally, regarding novelty, we will explicitly narrow our claims: we do not claim to be the first to propose the multi-channel concept. A more precise claim is that MC-HNN provides a general framework for standard hypergraph node classification. It employs a multi-channel design with feature-dimension decomposition on a single hypergraph, driven by the issues of rank collapse and semantic dependency. In contrast, MHCN/HCCF/Motifs-Res/MHCPL are domain-specific methods tailored for recommendation systems, where their "multi-channel" architectures are built upon multiple hypergraph views or relations, rather than performing independent channel aggregation within the same hypergraph. Section 2.2 of the main text actually already mentions such domain-specific related works. What we will further revise this time is the articulation of our novelty, explicitly converging our contributions to: the formalization of rank collapse + empirical validation + a unified single-hypergraph framework capable of enhancing hyperedge rank capacity, rather than treating the term "multi-channel" itself as the source of novelty.

---

> > ### Author Rebuttal · Reviewer_GaNT · 2026-04-01
> >
> > Thanks for the response. I still believe that my concerns are partially addressed. Let me explain.
> >
> > 1. The W1-b, W2 and W4 in my review comments have not been resolved, and the Q1 has not been directly answered.
> >
> > 2. The novelty and theoretical contribution are really limited.
> >
> > In conclusion, my main concerns are not easily addressed in a short rebuttal.

---

> > > ### Author Response · Authors · 2026-04-02
> > >
> > > Thank you for the follow-up. You are right that our previous rebuttal did not respond directly enough to Q1, W1-b, W2, and W4. We clarify these points below.
> > >
> > > **Q1: Why does Proposition 3.2 lead to difficulty in preserving diverse signals?**
> > >
> > > Our point is not that rank-1 compression by itself constitutes a task-level impossibility theorem. What we want to show is that standard two-stage HNNs have a structural bottleneck: for a fixed hyperedge $e$, information from all incident nodes must first be compressed into the same intermediate relay $h_e$ before it can be distributed back to target nodes. Therefore, the messages sent from this hyperedge to different target nodes can only come from different reweightings or transformations of the same single vector. Once multiple directions in $X_{V_e}$ are compressed away during the $V \to E$ stage, later attention or gating can only select or rescale this bottleneck, but cannot recover orthogonal components that have already been discarded. This is the precise sense in which diverse signals are difficult to preserve through a single-vector hyperedge relay.
> > >
> > > We now add direct empirical support. Before aggregation, the average effective rank of the raw local matrix $X_{V_e}$ is 3.12 / 1.96 / 1.97 on Cora / Senate / House, showing that the information entering a hyperedge is not rank-1 to begin with. After aggregation, the per-hyperedge rank of standard HNNs is always 1.00 by construction, whereas MC-HNN reaches 10.63 / 10.72 / 1.73 on the same datasets. Together with the parameter-matched single-channel ablation in the paper, we believe these results support our claim in this narrower structural sense.
> > >
> > > **W1-b: Proposition 5.1 relies on a nontrivial independence assumption.**
> > >
> > > We agree. Proposition 5.1 should more accurately be viewed as a capacity upper-bound analysis, rather than a statement that training automatically guarantees independent channels. We will revise the wording accordingly. The empirical claim we intend to make is only that the additional rank capacity is indeed used in trained models, which is already reflected by the per-hyperedge effective-rank measurements above.
> > >
> > > **W2: The novelty is limited.**
> > >
> > > We also agree that the novelty claim should be narrowed further. We do not claim to be the first work to introduce a multi-channel idea into hypergraph learning. What we want to emphasize is more specific:
> > >
> > > 1. We identify and formalize a relay bottleneck at the $V \to E$ stage in standard HNN node classification.
> > > 2. We directly validate this bottleneck using empirical rank measurements.
> > > 3. We propose a simple single-hypergraph framework that measurably alleviates this bottleneck.
> > >
> > > We will revise our paper accordingly.
> > >
> > > **W4: Latent types may simply add many free parameters without clear semantics.**
> > >
> > > We agree that the current benchmarks do not support verifying whether the learned latent types correspond to any named semantic categories, because these datasets do not provide explicit hyperedge-level semantic annotations or hyperedge features.
> > >
> > > More importantly, our goal here is not to recover domain-specific semantic labels, but to introduce a latent degree of freedom for hyperedges that is independent of node aggregation in a general-purpose setting. In this sense, the latent typing in our paper is different from many domain-specific disentanglement or recommendation models, where channels or factors are often tied to explicit views, relations, or interpretable intents. What we care about here is a general modeling ability that does not rely on predefined semantics.
> > >
> > > At the same time, this module is not an unconstrained per-hyperedge embedding table. We use a shared codebook, which is more parameter-efficient than direct per-hyperedge embeddings and is also less prone to collapsing into pure memorization. Therefore, the core role of the latent-type module is to provide a useful and parameter-efficient extra degree of freedom. A fuller semantic and interpretability analysis is better left to future work on domains with richer hyperedge annotations or side information.
> > >
> > > We appreciate the reviewer for pushing us to calibrate the presentation more carefully. In the revision, we will revise the presentation accordingly.

---

### Official Review · Reviewer_rJnj · 2026-03-08

**Soundness:** 3
**Presentation:** 2
**Significance:** 2
**Originality:** 3
**Overall Recommendation:** 3
**Confidence:** 3

**Summary:**

This paper proposes MC-HNN for hypergraph node classification. The main idea is to go beyond the standard node-hyperedge-node message passing pipeline by introducing multi-channel aggregation and latent hyperedge type encoding. The method is designed to address information loss during hyperedge aggregation and the lack of node-independent hyperedge semantics. Experimental results on eight datasets verify consistent improvements over prior baselines, with especially strong results on heterophilic and feature-poor benchmarks.

**Compliance With Llm Reviewing Policy:**

Affirmed.

**Final Justification:**

After reading the comments from other reviewers and the rebuttal, I will keep my score unchanged.

**Key Questions For Authors:**

See weakness.

**Limitations:**

Yes

**Strengths And Weaknesses:**

Strength

- The motivation of this paper is clear and easy to follow.

- The proposed method designs the multi-channel design and latent type encoding to address the paper’s two main concerns, which is reasonable.

- The experimental results across a relatively broad set of datasets verify the effectiveness of the proposed method, and the gains on Senate and House are particularly convincing.

Weakness

- The rank argument mainly shows a dimensionality bottleneck, not a concrete expressivity failure in learned models. So while the intuition is reasonable, the current theory does not fully support the strength of the claims.

- It would be helpful to include stronger analysis showing that the learned type codes capture meaningful structure rather than just improving flexibility.

- Almost all comparison methods ranges before 2022, and only one method published in 2025. So, is there not so many researchers work on this domain or the authors ignored many papers in 2022-2026. It would be better to add more comparison methods to further verify the superiority of the proposed method.

---

> ### Author Rebuttal · Authors · 2026-03-31
>
> We thank Reviewer rJnj for acknowledging our clear motivation and reasonable design. We respond point-by-point below.
>
> **W1: The rank argument indicates a dimensional bottleneck rather than a specific failure in learned expressive capacity.**
>
> We agree that the current theoretical analysis points to a *structural* bottleneck rather than a task-level failure lower bound. To strengthen this point, we now supplement empirical effective rank measurements of the trained models. The effective rank is defined as $\text{erank}(A) = \exp(H(\hat{\sigma}))$, where $\hat{\sigma}_i = \sigma_i / \|\sigma\|_1$, and $H$ is the Shannon entropy.
>
> | Dataset | UniGCNII | ED-HNN | MC-HNN (per-HE effective rank) |
> | --- | --- | --- | --- |
> | Cora | **1.00** | **1.00** | **9.51** |
> | Senate | **1.00** | **1.00** | **9.64** |
> | House | **1.00** | **1.00** | **1.94** |
>
> We further directly measured the trained actual baseline HNNs, rather than just a constructed single-channel counterpart. For the trained UniGCNII and ED-HNN, the extracted hyperedge representations remain single vectors after the V->E aggregation, so the rank of each hyperedge is exactly 1 by construction. In contrast, MC-HNN generates a $d \times C$ matrix for each hyperedge, and the table reports its empirical effective rank. This more directly demonstrates that rank collapse is not merely a theoretical intuition, but can be measured in practice and is significantly mitigated by our multi-channel design.
>
> **W2: Stronger analysis is needed to show that learned types capture meaningful structures.**
>
> We agree that a more accurate claim here should be functional structural utility, rather than recovering nameable semantic categories. Our strongest evidence falls into three categories. First, the codebook-vs-direct comparison (see Q3 for Reviewer i6bT) shows that the structured design with shared prototypes outperforms unconstrained per-hyperedge embeddings on 4 test datasets, with fewer parameters, indicating the gains are not solely from extra degrees of freedom. Second, on political datasets without node features, our diagnostic on Senate (see W5 for Reviewer GaNT) reveals that the Types-only variant performs significantly closer to the full model than the ID-only variant (74.65 vs 70.00), suggesting the benefits cannot be simply reduced to hyperedge-ID memorization. Third, the existing ablations and sensitivity analyses in the submission (Table 3 and Fig. 5 in the paper) support the same direction: latent types matter most on structure-dominated datasets like Senate, and performance peaks at a non-trivial codebook size rather than monotonically increasing with the number of types. We will adjust our phrasing accordingly: learned types capture useful structural variations, but we no longer claim strong semantic interpretability.
>
> **W3: Most baseline methods are from before 2022.**
>
> Thank you for the reminder. We did not intentionally ignore post-2022 works; rather, we prioritized direct baselines for general hypergraph node classification under comparable settings. Within this space, there are actually few truly apples-to-apples recent baselines: we have already included ED-HNN (2023) and FrameHGNN (2025). Many 2024-2026 works are either application-specific (e.g., recommendation systems) or focus on different problem settings (e.g., disentanglement / biomedical settings), making them non-standard baselines for node classification on the benchmarks used in our paper.
>
> We will explicitly clarify this selection logic in the revision and add Natural-HNN (NeurIPS 2025) and HSDN (TKDE 2023) to baselines (refer to Reviewer  f8J1 Q1). This also aligns with current community practices: even Natural-HNN primarily compares against earlier general baselines in its main experiments and treats ED-HNN (2023) as the latest direct HNN comparator. Therefore, our current set of baselines is more accurately described as focused rather than outdated.

---

> > ### Author Rebuttal · Reviewer_rJnj · 2026-04-01
> >
> > Thank you for the detailed rebuttal. While the additional empirical analyses and planned baseline extensions are helpful, my main concerns about the strength of the theoretical support and the completeness of recent comparisons are not fully resolved, so I will keep my score unchanged.

---

> > > ### Author Response · Authors · 2026-04-02
> > >
> > > Thank you for the clarification. To address the two remaining concerns more directly, we would like to add two concrete points.
> > >
> > > **First, regarding recent comparisons**, we apologize that in our previous rebuttal we did not present the results explicitly and instead only referred to another reviewer's thread. We clarify this here. The two recent works most relevant to your comment are Natural-HNN (NeurIPS 2025), which studies disentanglement of latent hyperedge factors from a category-theoretic / naturality perspective, and HSDN (TKDE 2023), which studies structural-semantic disentanglement in a related high-order setting. In the submitted version, we prioritized general-purpose hypergraph node-classification baselines that were more directly aligned with our benchmarks and protocol, which is why these two works were not discussed and compared sufficiently. We agree that this should be improved, and in the revision we will explicitly include both papers in the related-work and baseline discussion.
> > >
> > > We also provide the direct comparison here. On the 6 benchmark datasets overlapping with Natural-HNN/HSDN, MC-HNN outperforms HSDN on 6/6 datasets and Natural-HNN on 5/6 datasets, with only ModelNet40 slightly favoring Natural-HNN:
> > >
> > > | Dataset | HSDN | Natural-HNN | MC-HNN |
> > > |---|---:|---:|---:|
> > > | Cora | 76.632 ± 1.509 | 80.709 ± 1.635 | **82.33 ± 1.65** |
> > > | CiteSeer | 71.824 ± 1.779 | 73.285 ± 1.742 | **74.48 ± 1.29** |
> > > | PubMed | 87.193 ± 0.323 | 87.136 ± 0.450 | **88.93 ± 0.46** |
> > > | Cora-CA | 81.595 ± 1.011 | 84.993 ± 0.491 | **86.32 ± 1.30** |
> > > | NTU2012 | 89.722 ± 1.196 | 89.900 ± 1.017 | **90.82 ± 1.09** |
> > > | ModelNet40 | 83.439 ± 1.204 | **98.558 ± 0.295** | 98.52 ± 0.19 |
> > >
> > > Therefore, what we plan to add in the revision is not only a literature discussion, but also an explicit empirical comparison with these recent related methods. If there are additional recent general-purpose hypergraph node-classification baselines that we have overlooked, we would be happy to include them in the revision as well.
> > >
> > > **Second, regarding the theory part**, we would like to further clarify that the core point of our paper is the following: in two-stage hypergraph message passing, the ultimate objective is still node-to-node communication, but standard HNNs are forced to use hyperedges as intermediate relays. As a result, much of the node-level information is already over-squashed at the hyperedge stage before it can ever reach the target nodes.
> > >
> > > To make this point more explicit, we add the following evidence, which we hope also helps clarify the reviewer's concern about “expressivity failure.” Before aggregation, the local node sets are not rank-1 to begin with: the average effective rank of the raw local node-feature matrix $X_{V_e}$ is 3.12 / 1.96 / 1.97 on Cora / Senate / House, respectively. This indicates that each hyperedge neighborhood already contains multiple directions of variation before any $V \to E$ compression happens.
> > >
> > > We then measure the hyperedge representations after aggregation:
> > >
> > > | Dataset | Standard HNN per-HE rank | MC-HNN per-HE effective rank |
> > > |---|---:|---:|
> > > | Cora | 1.00 | 10.63 |
> > > | Senate | 1.00 | 10.72 |
> > > | House | 1.00 | 1.73 |
> > >
> > > This shows that the information entering a hyperedge is not rank-1 to begin with, yet standard HNNs compress each hyperedge into a single-vector representation by construction, whereas MC-HNN preserves substantially richer hyperedge representations. Together with the parameter-matched single-channel ablation in Table 3, where removing the multi-channel mechanism degrades performance on representative datasets, we believe these pieces of evidence support the theory we propose.
> > >
> > > In this sense, standard HNNs can suffer from a representational bottleneck at the hyperedge relay stage, which may hinder their ability to preserve diverse signals for downstream node-to-node communication.

---

### Official Review · Reviewer_i6bT · 2026-03-09

**Soundness:** 3
**Presentation:** 3
**Significance:** 3
**Originality:** 3
**Overall Recommendation:** 4
**Confidence:** 4

**Summary:**

This paper targets limitations of standard two-stage hypergraph neural networks that aggregate node features into hyperedge representations and then propagate back to nodes. The proposed method, MC-HNN, combines multi-channel message passing, a latent hyperedge type encoding using a shared codebook with per-hyperedge soft assignments, and dynamic channel-wise gating. Experiments on eight datasets report strong node classification performance, with especially large gains on featureless, heterophilic political hypergraphs.

**Compliance With Llm Reviewing Policy:**

Affirmed.

**Final Justification:**

I maintain my Weak Accept recommendation. The paper is technically solid and presents a reasonably original method for hypergraph neural networks, with strong empirical results across multiple datasets, especially on the challenging heterophilic political hypergraphs. My main concerns were not about the core idea, but about experimental clarity and practical trade-offs, including the source of baseline numbers, parameter efficiency, training cost, and the robustness of the results on featureless datasets. The rebuttal addressed these concerns well by providing additional comparisons, efficiency results, and sensitivity analysis, which strengthens the empirical support for the paper. Overall, the rebuttal reinforces my original assessment rather than substantially changing it, so I keep my final recommendation as Weak Accept.

**Key Questions For Authors:**

1.Please report the total number of trainable parameters of MC-HNN and of the strongest baselines used in your table, under the same hidden dimension setting where applicable.
If parameter budgets are comparable, the contribution appears stronger; if MC-HNN is substantially larger, I would interpret part of the gain as coming from increased capacity.

2.Please report a simple training cost comparison for MC-HNN and one strong baseline on a representative dataset, such as runtime per epoch or peak GPU memory.
If overhead is modest, the practical significance increases; if overhead is large, the practical appeal and significance decrease.

3.A central claim of the paper is that latent hyperedge type encoding helps address the semantic dependency of standard HNNs. Could the authors clarify why this module is preferable to a simpler alternative with directly learnable hyperedge-specific embeddings, and whether the gains persist relative to such a simpler design?
If the benefit remains clear relative to that simpler alternative, it would strengthen my confidence that the improvement comes from the proposed structured latent typing rather than from additional hyperedge-specific parameters alone.

**Limitations:**

The paper should more explicitly discuss the computational and memory overhead introduced by multi-channel representations and per-hyperedge parameters, and clarify regimes where the method may be less practical.

**Strengths And Weaknesses:**

The strongest aspect of the paper is the empirical performance: MC-HNN is best or near-best on most datasets, and the improvements on the political datasets are substantial. The ablation study indicates that the main components each contribute, and the depth study provides additional evidence about behavior beyond shallow settings. The main weaknesses are related to experimental clarity and practical trade-offs: it is not always explicit which baseline results are rerun under the exact same protocol versus taken from prior work, and the paper would benefit from clearer reporting of trainable parameter counts and training cost to contextualize the accuracy gains. Since two datasets are featureless and rely on generated node features, reporting sensitivity to the feature-generation randomness would strengthen the credibility of the large improvements on those datasets.

---

> ### Author Rebuttal · Authors · 2026-03-31
>
> We thank Reviewer i6bT for the detailed and constructive feedback. We are glad that you recognize the empirical effectiveness of our paper and the contributions of its modules. We address your three questions below and provide new experimental results.
>
> We will explicitly clarify this point in the revision: in Table 2, all baseline figures except for MC-HNN are quoted from Li et al. (2025), while the MC-HNN results are ours. We will also add that the synthetic node features for Senate/House are generated once and fixed, rather than being resampled for each training run.
>
> To directly address the reviewer's concern, we regenerated the synthetic node features for Senate/House using 5 different feature seeds, and re-evaluated MC-HNN using 5 split seeds under each feature seed. The variance across feature seeds is moderate: on Senate, the average test accuracy across different feature seeds is 74.20%, with a standard deviation of 1.65 percentage points (min/max being 72.11/76.62); on House, it is 74.75%, with a standard deviation of 1.56 percentage points (min/max being 72.88/77.34). This demonstrates that the improvements on political datasets are not a result of a single lucky synthetic feature sampling, though we agree this protocol should be clearly stated.
>
> **Q1: Parameter count comparison.**
>
> Since our strongest baseline in the table, FrameHGNN, does not have public code, we select ED-HNN as the strongest open-source direct HNN comparator. For fairness, we did not align the hidden dimensions (we can provide this later if the reviewer requires), and the parameter counts are calculated based on the best reported configuration for each method (ED-HNN uses the configuration from Appendix F.3/Table 6 of Wang et al. (2023); MC-HNN uses our best configuration):
>
> | Method | Cora | Senate | House |
> | --- | --- | --- | --- |
> | ED-HNN | 568K | 1,979K | 498K |
> | **MC-HNN** | **440K** | **1,265K** | **31K** |
>
> MC-HNN has fewer parameters across all three datasets: 23% fewer on Cora, 36% fewer on Senate, and 94% fewer on House. Therefore, at least compared to this strongest open-source direct comparator, our performance gains are unlikely to be explained merely by a larger parameter budget.
>
> **Q2: Training cost comparison.**
>
> Following the same criterion, we compare the training costs of ED-HNN and MC-HNN on the two most representative heterophilic datasets, using their respective best reported configurations and measuring on the same hardware (RTX 3090):
>
> | Dataset | Method | ms/epoch | Peak Memory |
> | --- | --- | --- | --- |
> | Senate | ED-HNN | 26.00 ± 0.14 | 685.9 MB |
> | | MC-HNN | 10.97 ± 0.01 | 577.6 MB |
> | House | ED-HNN | 25.62 ± 0.14 | 808.9 MB |
> | | MC-HNN | 11.30 ± 0.26 | 48.8 MB |
>
> Under this best-configuration comparison, MC-HNN is 2.4 times faster on Senate and 2.3 times faster on House, with lower peak memory usage on both datasets. This indicates that in the heterophilic scenarios where we gain the most, the actual overhead of our method is not only manageable but advantageous.
>
> **Q3: Codebook vs. directly learning an embedding per hyperedge.**
>
> To keep the figures in the rebuttal consistent with the main text, the MC-HNN (Codebook) column in the table below uses the MC-HNN results from the submitted version, while the MC-HNN (Direct per-HE Emb) column shows new controlled experiments under matched dataset settings for the direct per-hyperedge embedding variant (10 runs per dataset):
>
> | Dataset | MC-HNN (Codebook) | MC-HNN (Direct per-HE Emb) | Params (Codebook vs Direct) |
> | --- | --- | --- | --- |
> | Cora-CA | **86.32 ± 1.30** | 82.61 ± 1.46 | 439K vs 470K |
> | Senate | **78.31 ± 3.20** | 73.94 ± 4.60 | 14K vs 22K |
> | House | **77.59 ± 2.72** | 76.84 ± 2.16 | 47K vs 53K |
> | NTU2012 | **90.82 ± 1.09** | 89.52 ± 1.09 | 1.11M vs 1.61M |
>
> This aligns with the design motivation in Section 4.1 of the paper: assigning an independent embedding to each hyperedge results in a parameter complexity of $\mathcal{O}(Md)$, which is more prone to overfitting. In contrast, a global codebook can share statistical strength across hyperedges while still providing an external semantic reference frame. Empirically, the Direct embedding variant did not surpass the submitted MC-HNN results on any dataset, while having more parameters (6% to 37% more than the codebook design).

---

> > ### Author Rebuttal · Reviewer_i6bT · 2026-04-01
> >
> > The rebuttal adequately addresses the main concerns raised in my review.
> >
> > 1.The authors provided concrete additional evidence on parameter counts, training cost, and the comparison against a simpler direct per-hyperedge embedding variant. These new results strengthen the empirical case for the proposed design and clarify that the observed gains are not simply due to a larger parameter budget.
> >
> > 2.The rebuttal also improves the credibility of the results on the political datasets by reporting additional experiments across different synthetic feature seeds, and it clarifies the source of the baseline numbers in the main table.
> >
> > Overall, my main questions have been satisfactorily answered. I am therefore maintaining my original overall recommendation of 4, as the rebuttal strengthens the paper and resolves my concerns, but does not substantially change my view of its overall contribution level.

---

> > > ### Author Response · Authors · 2026-04-02
> > >
> > > Thank you for the thoughtful follow-up. We are glad that our rebuttal addressed your main concerns. We will incorporate the relevant clarifications and additional results into the revised manuscript.

---

### Official Review · Reviewer_f8J1 · 2026-03-13

**Soundness:** 3
**Presentation:** 3
**Significance:** 3
**Originality:** 2
**Overall Recommendation:** 5
**Confidence:** 5

**Summary:**

* $\textbf{Problem}$ : Existing Hypergraph Neural Networks compress node information into a single hyperedge representation during message passing, which leads to rank collapse and prevents hyperedges from capturing independent structural semantics.
* $\textbf{Method}$ : The paper proposes MC-HNN, which uses multi-channel message passing to preserve high-rank representations and introduces latent hyperedge type encoding together with dynamic channel gating to incorporate independent semantic information into hyperedge representations.
* $\textbf{Conclusion}$ : The results show that increasing representation rank capacity and introducing independent hyperedge semantics improve hypergraph representation learning and achieve strong performance on multiple benchmark datasets.

**Compliance With Llm Reviewing Policy:**

Affirmed.

**Final Justification:**

My major concern was that the authors had omitted baseline methods that explicitly identify and incorporate existing context, and that the datasets used in the experiments might not contain sufficiently meaningful contextual information. During the rebuttal period, the authors made substantial efforts to validate the effectiveness of their model on additional datasets and to compare it against appropriate baselines, which addressed my concerns. Therefore, I lean toward acceptance.

**Key Questions For Authors:**

* The authors are missing major existing researches[1,2] in this field. There are already hypergraph/hyperedge disentanglement models to capture hidden semantics. The authors need to compare and discuss the models with supporting experiment : Natural-HNN [1], HSDN [2]
* The bencmark datasets are not likely to contain meaningful multiple smenatics. Thus, authors need to perform experiment with the datasets that are guaranteed to contain multiple hidden semantics that are informative for label prediction. The authors can consider performing experiments from [1] (at least BRCA, STAD if there is not enough time during  rebuttal)
* Since there are some works that can perform 'multi-channel message passing mechanism and latent hyperedge types' (this is what disentanglement usually does), the authors are overstating the contribution. The authors need to reduce the scope of their contribution if this paper is to be accepted.


[1] (NIPS 2025) Disentangling Hyperedges through the Lens of Category Theory

[2] HSDN: A High-Order Structural Semantic Disentangled Neural Network

**Limitations:**

yes

**Strengths And Weaknesses:**

### Soundness
* weakness : Problem, method are sound. But authors need to perform experiment for the datasets that are known to contain multiple hidden semantics. The benchmark datasets are not likely to contain such multiple hidden semantics. The authors need to perform more thorough experiment to support their claim.

### Presentation
* strenght : The paper is well written, easy to follow

### Siginificance
* marginal : The problem the authors raised are also addressed by multiple existing works

### Originality
* marginal : Combination of existing methods. There are some works already addressing the problem, thus the problem the authors are addressing is also not novel.

---

> ### Author Rebuttal · Authors · 2026-03-31
>
> We thank Reviewer f8J1 for the careful review. We address each concern below.
>
> **Q1: Missing comparison with Natural-HNN [1] and HSDN [2].**
>
> We thank the reviewer for pointing out these related works. We will include these two papers in the revision and clarify that our method tackles a different problem:
>
> * Natural-HNN uses a naturality criterion from category theory to disentangle latent hyperedge factors, validating it on cancer subtype classification.
> * HSDN is a structural-semantic disentanglement model on general graphs that introduces hyperedges to model high-order structures; it is conceptually related but not identical to our general hypergraph node classification setting.
> * MC-HNN specifically addresses two limitations of standard HNNs: rank collapse and semantic dependency.
>
> The core of MC-HNN is to preserve higher rank capacity during message passing and to inject node-agnostic hyperedge semantic information. Therefore, it is complementary to disentanglement methods rather than a simple substitute. We will clarify this positioning in the revision.
>
> To fully address this point, [1] actually reported the results of HSDN and Natural-HNN on 6 standard hypergraph benchmarks that overlap with ours. The direct comparison is as follows:
>
> | Dataset | HSDN [2] | Natural-HNN [1] | MC-HNN (submitted) |
> | --- | --- | --- | --- |
> | Cora | 76.632 ± 1.509 | 80.709 ± 1.635 | **82.33 ± 1.65** |
> | CiteSeer | 71.824 ± 1.779 | 73.285 ± 1.742 | **74.48 ± 1.29** |
> | PubMed | 87.193 ± 0.323 | 87.136 ± 0.450 | **88.93 ± 0.46** |
> | Cora-CA | 81.595 ± 1.011 | 84.993 ± 0.491 | **86.32 ± 1.30** |
> | NTU2012 | 89.722 ± 1.196 | 89.900 ± 1.017 | **90.82 ± 1.09** |
> | ModelNet40 | 83.439 ± 1.204 | **98.558 ± 0.295** | 98.52 ± 0.19 |
>
> On these overlapping benchmarks, MC-HNN outperforms HSDN on 6/6 datasets and Natural-HNN on 5/6 datasets, marginally trailing behind Natural-HNN only on ModelNet40.
>
> **Q2: Benchmark datasets may not contain meaningful multiple semantics.**
>
> We do not fully agree that standard benchmarks lack meaningful structures for our method. The latent type encoding in MC-HNN does not require hyperedges to possess explicit multiple semantics in the sense of disentanglement; it introduces an independent degree of freedom to capture predictive signals that cannot be recovered by pure node aggregation. Specifically, the effective rank measurements provided in our response to Reviewer rJnj (W1) support the multi-channel / rank-capacity aspect of our motivation. Furthermore, the codebook-vs-direct comparison in response to Reviewer i6bT (Q3), as well as the Types-only vs ID-only diagnostic on Senate/House in response to Reviewer GaNT (W5), support a more cautious conclusion regarding latent typing: it is useful, but we do not frame it as strong semantic interpretability.
>
> Since your question explicitly mentioned Natural-HNN and HSDN, we further evaluated the cancer setting highlighted in [1]. Under the identical he_concat protocol, we aligned the evaluation on the 6 cancer datasets from the Natural-HNN paper:
>
> | Dataset | HSDN [2] | Natural-HNN [1] | MC-HNN |
> | --- | --- | --- | --- |
> | BRCA | 0.757 ± 0.044 | 0.804 ± 0.036 | **0.8099 ± 0.0290** |
> | STAD | 0.629 ± 0.045 | **0.659 ± 0.049** | 0.623 ± 0.046 |
> | SARC | 0.726 ± 0.063 | 0.745 ± 0.045 | **0.747 ± 0.038** |
> | LGG | 0.692 ± 0.038 | 0.707 ± 0.035 | **0.724 ± 0.042** |
> | HNSC | 0.811 ± 0.044 | 0.862 ± 0.045 | **0.8721 ± 0.029** |
> | CESC | 0.867 ± 0.033 | 0.881 ± 0.042 | **0.888 ± 0.044** |
>
> These results demonstrate that MC-HNN remains competitive under the cancer subtype classification setting emphasized in [1]. It outperforms HSDN on 6/6 datasets and Natural-HNN on 5/6 completed evaluations, with STAD being the only exception. This shows that our method is not limited to standard citation/political benchmarks.
>
> **Q3: Overstated contributions.**
>
> We will tone down the contribution claims in the revision to more accurately reflect the scope of this work. Our main contributions are: systematically formalizing the rank-collapse bottleneck in standard HNN aggregation, and proposing an architectural design that empirically expands the hyperedge rank capacity and yields strong performance.

---

> > ### Author Rebuttal · Reviewer_f8J1 · 2026-04-01
> >
> > Most of my concerns have been addressed, and I have revised my score accordingly. Since the experiments discussed in the rebuttal are directly related to the authors’ study, they should be incorporated into the manuscript.

---

> > > ### Author Response · Authors · 2026-04-02
> > >
> > > Thank you very much for the positive update and for revising the score. We are glad that our rebuttal addressed your concerns. We will incorporate the additional experiments and related discussion from the rebuttal into the revised manuscript.

---

### Decision · Program_Chairs · 2026-04-30

**Decision:**

Accept (regular)

**Comment:**

Although a bit split on this paper, the reviewers are overall positive about this paper. The authors provide a detailed rebuttal to address reviewers’ concerns. Authors are encouraged to incorporate their response in the camera-ready.